# SchemaRAG: Enhancing Knowledge-Intensive Reasoning of LLMs via Inference-Time Adaptive Schema

## Abstract

Retrieval-Augmented Generation (RAG) often struggles with integrating fragmented knowledge for complex reasoning tasks. Recent efforts introduce structural templates—such as graphs or knowledge-based organizations—to improve multi-document reasoning. However, they are constrained by their rigidity, failing to adapt to diverse, task-specific information structures and often omitting critical dependencies. To address this, we propose SchemaRAG: an adaptive schema-guided RAG framework. Instead of predefined formats like graphs, tables and chunks, SchemaRAG adaptively organize the factual information across documents based on query-specific requirements. Given the input query and documents, it first parses the query into sub-problems and generate strategies for schema constructions, then utilize the organized knowledge to generate final answer. Extensive experiments on real-world benchmarks demonstrate that SchemaRAG consistently outperforms state-of-the-art baselines in knowledge-intensive reasoning and generation quality. Our work highlights the importance of adaptive schema-guided strategies for advancing the capabilities of RAG systems in complex, domain-specific tasks.

## 1 Introduction

Large Language Models (LLMs) (Zhao et al. (2023)) have demonstrated remarkable capabilities across diverse real-world tasks. However, their performance in specialized domains is often constrained by the scope and coverage of their training data (He et al. (2022); Jiang et al. (2023)). Retrieval-Augmented Generation (RAG) methods (Guu et al. (2020); Lewis et al. (2020)) seek to overcome these knowledge limitations by retrieving semantically relevant external information and dynamically incorporating it into the generation process. Nevertheless, they often struggle to capture global relationships among fragmented pieces of information, resulting in suboptimal performance on reasoning tasks that require integrating knowledge from multiple documents (Edge et al. (2024)).

To address these limitations, structure-enhanced RAG methods (He et al. (2024); Li et al. (2025b)) have been developed, categorized into three main streams: **1) Index-based Methods** (Edge et al. (2024); Guo et al. (2024); Jimenez Gutierrez et al. (2024)) utilize structural representations to model complex relationships among textual units, enabling a more comprehensive understanding across multiple-source information. For instance, GraphRAG (Edge et al. (2024); Wu et al. (2024a)) constructs graph indices over external corpora to generate community-based summaries. **2) Knowledge-based Methods** (Luo et al. (2023); Sun et al.; Panda et al. (2024); Chen et al. (2024b)) leverage structural information organization to shorten reasoning paths and enhance generation quality. Approaches such as SGP (Cheng et al. (2024)) and StructRAG (Li et al. (2025b)) employ task-agnostic prompting to choose a specific structural form during inference.

Despite their effectiveness, existing methods are mainly constrained by reliance on single structural format (e.g., graph) and fixed templates, limiting adaptability to diverse domain-specific structural requirements, as illustrated in Figure 1. This rigidity often results in the omission of critical task-specific information or the inclusion of irrelevant content during reorganization, thereby impeding the accurate identification of key metrics and distinctions essential for effective task performance. Table 1 provides a comprehensive comparison of existing structure-based RAG methods, highlight-

Figure 1: Comparison of the existing structure-based RAG and adaptive schema RAG: Unlike predefined formats and templates, SchemaRAG dynamically constructs knowledge schemas using an adaptive strategy. This approach tailors the schema to query-specific tasks and domains, enabling more flexible and efficient integration with external knowledge and LLMs.

Table 1: Comprehensive comparisons with existing structure-enhanced RAG methods. The categories include KG-RAG and Doc-RAG, representing the paradigms of using pre-built KG and original corpus documents as inputs, respectively. "Selection" denotes the choice of structure types. The structured objects consist of index, knowledge, and task, denoted as I, K, and T, respectively.

| Method | Category | Structured Objects | Non-Single Structures | Composite Structures | Adaptive Selection | Adaptive Construction | Adaptive Utilization |
|---|---|---|---|---|---|---|---|
| G-retriever (He et al. (2024)) | KG-RAG | K | ✗ | ✗ | ✗ | ✗ | ✗ |
| RoG (Luo et al. (2023) ) | KG-RAG | K | ✗ | ✗ | ✗ | ✗ | ✓ |
| KnowTrace (Li et al. (2025a)) | Doc-RAG | K | ✗ | ✗ | ✗ | ✓ | ✗ |
| RAPTPOR (Sarthi et al. (2024)) | Doc-RAG | I/K | ✗ | ✓ | ✗ | ✗ | ✗ |
| LightRAG (Guo et al. (2024)) | Doc-RAG | I/K | ✗ | ✗ | ✗ | ✗ | ✗ |
| GraphRAG (Edge et al. (2024) ) | Doc-RAG | I/K | ✗ | ✓ | ✗ | ✗ | ✗ |
| StructRAG (Li et al. (2025b)) | Doc-RAG | K/T | ✓ | ✗ | ✓ | ✗ | ✗ |
| Ours | Doc-RAG | I/K/T | ✓ | ✓ | ✓ | ✓ | ✓ |

ing their respective strengths and limitations." For example, when dealing with judgment determination tasks in the legal domain, it is crucial to extract key clauses from multiple documents and organize the information according to the specific requirements of each case. However, fixed graph-based templates are typically limited to extracting and summarizing relevant triples, lacking the flexibility to adapt to evolving case-specific demands.

To overcome this, we draw inspiration from schema theory Widmayer (2004), which highlights how humans organize external information into schemas—hierarchical and interconnected structures—and flexibly adjust them to suit varying cognitive tasks. Unlike static structures that merely arrange information, human schemas enable flexible organization and utilization across diverse contexts. Inspired by this, we argue that existing RAG methods urgently needs incorporate inference-time "adaptive schemas". We define a knowledge schema as a adaptively constructed composite structure that organizes factual information based on task-specific requirements. A clearer definition is in the Sec.2. As shown in Figure 1, the adaptive schema RAG method adaptively aligns structural strategies with varying knowledge characteristics, addressing rigidity and improving generation performance. However, building this capability presents two main challenges: *C1: How to adaptively leverage external documents to organize suitable knowledge schema? C2: How to flexibly and effectively utilize knowledge schema to address varying complex tasks?*

To overcome these challenges, we proposes SchemaRAG, an adaptive schema-guided RAG method, which comprises three core components: schema thinking, schema construction, and schema utilization. To address *C1*: we introduce schema thinking to parse the characteristics of query and documents to generating schema strategy via step-by-step thinking. Building on this, schema construction leverages the identified suitable structure and construction strategy to adaptively assemble the knowledge schema. To address *C2*: in schema utilization phase, we design a hierarchical retrieval-merge mechanism that incrementally extracts relevant knowledge from the schema according to schema strategy and subquery dependencies, and ultimately merges the results to generate high-quality answers to the original query.

Our main contributions are as follows:

- We systematically analyze the critical limitations of existing structure-enhanced RAG methods and motivate the necessity for adaptive schema-guided approaches to address task-specific challenges.
- We propose SchemaRAG, an adaptive schema-guided RAG framework that adaptively constructs and utilizes the most suitable knowledge schema for varying specific tasks, substantially enhancing knowledge-intensive reasoning.

- We conduct comprehensive experiments on real-world benchmarks, demonstrating that SchemaRAG consistently outperforms state-of-the-art baselines in generation quality.

## 2 PRELIMINARIES

### 2.1 TASK FORMULATION

We formally define the knowledge-intensive reasoning task based on the following components:

- **Data:** The task considers an input question $Q$ and a collection of documents $\mathcal{D} = \{D^{(1)}, D^{(2)}, D^{(3)}, \ldots, D^{(n)}\}$, where $n$ can be substantial (e.g., $n > 20$), often resulting in a cumulative token count exceeding 200K. A characteristic of this setting is that relevant information is broadly dispersed across multiple documents.
- **Model:** The objective is to infer an answer $A$ by leveraging the distributed information in $\mathcal{D}$. We aim to devise a function $\mathcal{F}$ such that $A = \mathcal{F}(Q, \mathcal{D})$, where $\mathcal{F}$ encompasses not only the answer generation process but also the retrieval and organization of supporting evidence from the provided documents.

For instance, in the legal domain, when assessing the consistency of judicial decisions require extracting legal principles, case facts, and reasoning processes scattered across multiple judgment documents, followed by their comparison and synthesis while considering court hierarchies and case-specific contexts. Such knowledge-intensive reasoning tasks pose significant challenges in retrieving and integrating evidence from large-scale and heterogeneous document corpora. To this end, we define a novel structure form, named knowledge schema, to flexibly process the challenge in knowledge-intensive reasoning scene.

### 2.2 KNOWLEDGE SCHEMA

Formally, we provide a clear definition of the knowledge schema described in this paper.

**Definition 1 (Knowledge Schema)** We define a schema $\mathcal{S}$ as a hierarchical composite structure that encapsulates fact organization and their relationships. Specifically:

- **Knowledge-type Structure**: Each knowledge-type structure $U^{(i)}$ is constructed as a flexible aggregation of factual information units, formally defined as: $U^{(i)} = K_j(\{f_1^{(i)}, f_2^{(i)}, \ldots, f_k^{(i)}\})$, where each $f_k^{(i)}$ denotes an atomic fact or knowledge element extracted from document $D^{(i)}$, and $K_j(\cdot)$ represents a knowledge-level structuring operator that organizes these facts into a coherent structure. Here, $K_j \in \mathcal{K}$, where $\mathcal{K}$ denotes a set of candidate structuring operators, each parameterized or selected according to the context of the task, question, or domain.
- **Index-type Structure**: The index-type structure is defined as a higher-level structure that consists of a collection of knowledge-type structures $U^{(i)}$ from document $D^{(i)}$ and capture cross-document associations and provide efficient indexing among knowledge units. The ultimate representation of the knowledge schema $\mathcal{S}$: $\mathcal{S} = I_j(\{U^{(1)}, U^{(2)}, \ldots, U^{(n)}\})$, and $I_j(\cdot)$ represents a index-level structuring operator that organizes these units into a coherent structure. Here, $I_j \in \mathcal{I}$, where $\mathcal{I}$ denotes a set of candidate structuring operators.

## 3 METHODOLOGY

In this section, we introduce our proposed SchemaRAG, designed for knowledge-intensive reasoning tasks. As illustrated in Figure 2, the framework comprises three modules: schema thinking, schema construction, and schema utilization. Each component is described in detail below.

### 3.1 SCHEMA THINKING

Inspired by human intuition in processing new knowledge, we argue that constructing a knowledge schema should be guided by a thinking mechanism designed to analyze task requirements and the distribution of information in documents, gradually deriving the optimal schema organization and strategy. To this end, we propose a schema-based thinking paradigm, which integrates task structural

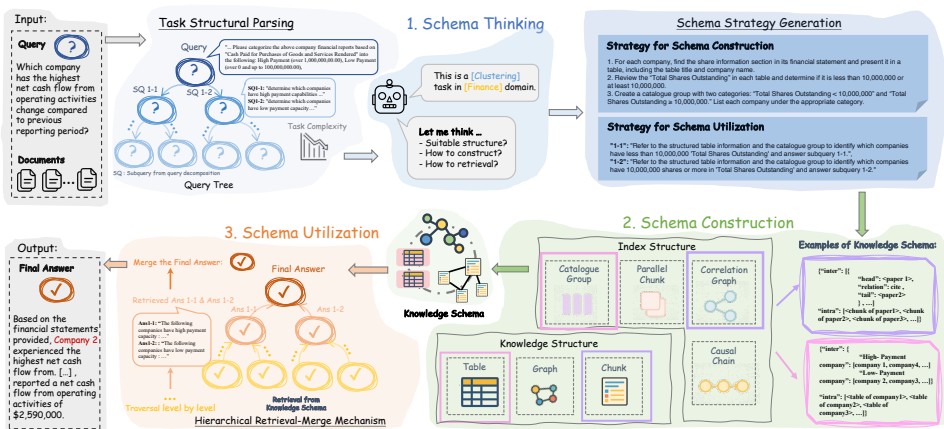

Figure 2: The overview of the proposed SchemaRAG. It begins with the Schema Thinking module, which parses the input query into sub-problems and formulates schema strategies based on task requirements. Subsequently, the Schema Construction module integrates domain knowledge and index structures to construct a comprehensive knowledge schema. Finally, the Schema Utilization module leverages a hierarchical retrieval-merge mechanism to iteratively gather and combine relevant information, producing the final answer to the query.

parsing and schema strategy generation. This approach enables the formulation of task-specific strategies for schema construction and their effective utilization.

**Task Structural Parsing.** To tackle complex tasks, we introduce a top-down structural parsing strategy that hierarchically decomposes a query into sub-problems, mirroring human step-by-step problem-solving. This approach forms a tree $T$, where each node $t = (q, l, s) \in T$ represents a sub-query $q$, its depth level $l$, and its state $s \in$ "continue","end". Starting from the root node (the initial query $Q$), if the state is "continue," an LLM-based parser $\mathcal{P}$ generates child sub-queries $(q_j^{(l+1)}, s_j^{(l+1)})$ based on a decomposition prompt: $\left\{(q_j^{(l+1)}, s_j^{(l+1)})\right\} = \mathcal{P}(q_i^{(l)})$. This iterative process continues until all leaf nodes reach the "end" state, producing a complete query tree $T$ that captures logical dependencies and intermediate reasoning steps. The final query tree $T$ can be represented as a dictionary: $T = \{$"Query" : $Q$,"Subquery(1-1)" : $q_1^{(1)}, \ldots,$"Subquery(l-i)" : $q_i^{(l)}, i = 1, \ldots, N\}$, where $q_i^{(l)}$ denotes the $i$-th sub-query at depth $l$, and $N$ is the total number of leaf nodes. This structured format enables systematic and interpretable solutions for complex reasoning tasks.

**Schema Strategy Generation.** Building upon the task structural parsing described above, we introduce a CoT-based method for schema strategy generation to guide the construction and utilization of knowledge schemas. We define the strategy generator $\mathcal{G}$ that orchestrates the following steps:

**Step 1: Task Analysis and Typing.** Given the query tree $T$ and the core contents $\mathcal{H}$ of the document collection $\mathcal{D}$, the generator assigns a task type $y \in \mathcal{Y}$ and performs task description $d$, including domain identification, for each instance:

$$y, d = \mathcal{G}(T, \mathcal{H}), \quad \text{where} \quad \mathcal{H} = \{h^{(i)}\}_{i=1}^{n}. \tag{1}$$

Here, $\mathcal{H}$ represents the titles or key sentences of each document $D$, and $\mathcal{Y}$ denotes a predefined set of candidate task types, including summarization, comparison, clustering, extraction, planning, and multi-hop reasoning.

**Step 2: Structure Combination.** Using the task information from step 1 and the example document $D_e$, the generator $\mathcal{G}$ selects an appropriate knowledge structure $K$ and index structure $I$ to construct the knowledge schema:

$$K, I = \mathcal{G}(y, d, D_e), \tag{2}$$

where $y$ and $d$ denote the task type and task description, respectively. Here, $K \in \mathcal{K}$ and $I \in \mathcal{I}$ represent the selected knowledge structure and index structure, respectively. The predefined candidate sets include knowledge structures $\mathcal{K}$: table, graph, and chunk; and index structures $\mathcal{I}$: causal chain, correlation graph, parallel chunk, and catalog group, denoting causal, relevant, irrelevant, and clustering relationships among documents. For instance, as shown in Figure 2, the pink wire-

frame indicates that the selected $K$ and $I$ are "Table" and "Catalogue Group", respectively. These structures can also be used to express the corresponding structural operators.

**Step 3: Strategy Generation.** Based on the previous thinking process, the generator is considered to have the ability to provide guidance for constructing schema forms with document information and for extracting subquery-relevant information from the schema. Given the full document collection $\mathcal{D}$ and the previous results, we prompt $\mathcal{G}$ to generate executable procedures $R_{con}$ and $R_{uti}$ as the strategies for schema construction and utilization, respectively.

$$R_{con}, R_{uti} = \mathcal{G}(T, \mathcal{D}, K, I). \tag{3}$$

Here, $R_{uti} = R_i^{(l)}$ contains operation guidance items that are sequentially generated for the subqueries in the query tree. $R_{con} = \{R_{con}^K, R_{con}^I\}$, where $R_{con}^K$ and $R_{con}^I$ denote the construction strategies for the knowledge structure and the index structure, respectively; these can be included in the prompt for structural operations.

## 3.2 SCHEMA CONSTRUCTION

Based on the preliminaries, our knowledge schema comprises a knowledge structure and an index structure, forming an adaptive knowledge organization framework. The construction process is detailed below.

**Knowledge Structure Construction.** Given the construction strategy for the knowledge-type structure $R_{con}^K$, we derive adaptive, task-specific prompts and employ a guided LLM as a knowledge-level structuring operator as follows:

$$K(\cdot) = \text{LLM}(\text{prompt} \| R_{con}^K, \cdot), \tag{4}$$

where prompt denotes a general, task-agnostic instruction, $\|$ represents the concatenation operation, and $(\cdot)$ indicates the input operator. We then organize the intra-document factual knowledge as knowledge units $U^{(i)}$ for each document $D^{(i)}$:

$$U^{(i)} = K(D^{(i)}). \tag{5}$$

**Index Structure Construction.** Given the construction strategy for the index-type structure $R_{con}^I$, we derive adaptive, task-specific prompts and employ a guided LLM as an index-level structuring operator as follows:

$$I(\cdot) = \text{LLM}(\text{prompt} \| R_{con}^I, \cdot), \tag{6}$$

where prompt denotes a general, task-agnostic instruction, $\|$ represents the concatenation operation, and $(\cdot)$ indicates the input operator.

Finally, by organizing intra-document information into knowledge structure units and integrating inter-document relationships as an index structure, we obtain the knowledge schema $\mathcal{S}$ as follows:

$$\mathcal{S} = I(\{U^{(1)}, U^{(2)}, \dots, U^{(n)}\}). \tag{7}$$

## 3.3 SCHEMA UTILIZATION

Building on the constructed knowledge schema, we enhance final answer generation through a flexible utilization strategy. Additionally, we introduce a hierarchical retrieval-merge mechanism guided by the query tree and the thinking phase strategy.

**Hierarchical Retrieval-Merge Mechanism.** Following the principle of solving problems from simple to complex, we propose a bottom-up hierarchical traversal process for the query tree $T$. At the $i$-th leaf node of the $l$-th level, the related knowledge $\hat{a}$ for the current subquery is retrieved from the knowledge schema as follows:

$$\hat{a}_i^{(l)} = \mathcal{Z}_{retrieval}(\mathcal{S}, q_i^{(l)}, R_i^{(l)}), \tag{8}$$

where $\mathcal{Z}_{retrieval}$ represents an LLM-based generator that operates under the utilization strategy $R_i^{(l)}$. Once the retrieval process at the $l$-th level is completed, a merging operation is performed.

Based on the subquery nodes at the current level and the retrieved knowledge $\hat{a}^{(l)}$, the answer $a$ for the parent node's corresponding subquery $q_j^{(l-1)}$ is obtained as:

$$a_j^{(l-1)} = \mathcal{Z}_{merge}(\hat{a}_j^{(l-1)}, \hat{a}_1^{(l)}, \ldots, \hat{a}_i^{(l)}). \tag{9}$$

When the traversal reaches the root node, corresponding to the original query $Q$, the final answer is derived as $Answer = a^{(0)}$.

This utilization mechanism incrementally retrieves relevant knowledge from the schema according to the schema strategy and subquery dependencies, and then merges the results to generate high-quality answers to the original query $Q$.

## 4 EXPERIMENTS

In this section, we conduct experiments to answer the following research questions:

- **RQ1**: How does generation performance of SchemaRAG compare to that of established RAG baselines?
- **RQ2**: What specific advantages does SchemaRAG demonstrate through case studies in various scenarios?
- **RQ3**: How do adaptive schema generation and hierarchical retrieval-merge mechanisms enhance output quality of SchemaRAG?
- **RQ4**: How do the resource costs of SchemaRAG compare to those of baseline approaches?

### 4.1 EXPERIMENTAL SETTINGS

**Datasets & Metrics.** To evaluate knowledge-intensive reasoning, we use the Loong benchmark (Wang et al. (2024)), which includes three domain-specific, document-based question answering tasks in legal, finance, and academic domains. It covers four reasoning types: spotlight locating, comparison, clustering, and multi-hop reasoning, with document lengths ranging from 10,000 to 250,000 tokens. Following the evaluation protocol in Wang et al. (2024), we assess SchemaRAG using LLM-judged scores (0–100) and exact match (EM) rates. To mitigate ambiguity in LLM-based scoring, we also analyze the correlation between human expert evaluations and LLM scores, as shown in Table 5 in the Appendix.

**Baselines & Details.** We select the common used and recent methods for knowledge-intensive question answering task as our compared baselines. Mainly includes the following five methods:

- **Long-Context**( Team (2024)): which is a foundational LLM capable of processing large-scale inputs and handling up to 128k tokens.
- **Naive RAG**( Lewis et al. (2020)): which segments documents into passages and uses dense retrieval to select relevant content for answer augmentation.
- **RQ-RAG**( Chan et al. (2024)): which utilizes a fine-tuned LLM to decompose the original query, thereby facilitating the retrieval of relevant document segments.
- **GraphRAG**( Edge et al. (2024)): which constructs graph communities over the original document and leverages the graph structure to capture cross-document relevant fragments.
- **RAPTOR**( Sarthi et al. (2024)): which introduces recursive embedding, clustering, and summarization of text chunks, constructing a bottom-up hierarchical summarization tree.
- **HIPPORAG2**( Gutiérrez et al.): which enhances the Personalized PageRank algorithm from HippoRAG with deeper passage integration and more efficient online LLM utilization.
- **StructRAG**( Li et al. (2025b)): which reorganizes information with various structural formats, leveraging structured text to enhance task performance.

To ensure fairness, we utilize Qwen2-72B-Instruct as base model and employ GPT-4o (2024-04-13) as the evaluation model for all methods, following the same setting as in Loong (Wang et al. (2024)).

### 4.2 OVERALL PERFORMANCE (RQ1)

To evaluate the effectiveness of our approach, we conduct comprehensive comparisons against a diverse set of baseline methods on the Loong dataset. The results are summarized in Table 2. We highlight the following key observations:

Table 2: Performance comparison on the Loong benchmark.

| Method | Spot. | | Comp. | | Clus. | | Chain. | | Overall | |
|---|---|---|---|---|---|---|---|---|---|---|
| | LLM Score | EM | LLM Score | EM | LLM Score | EM | LLM Score | EM | LLM Score | EM |
| **Set 1 (10K-50K Tokens)** | | | | | | | | | | |
| Long-context (Team (2024)) | 68.49 | **0.55** | 60.60 | 0.37 | 47.08 | 0.08 | 70.39 | 0.36 | 60.11 | 0.29 |
| Naive RAG (Lewis et al. (2020)) | 51.08 | 0.35 | 44.53 | 0.27 | 37.96 | 0.05 | 53.95 | 0.35 | 46.11 | 0.23 |
| RQ-RAG (Chan et al. (2024)) | **72.31** | 0.54 | 48.16 | 0.05 | 47.44 | 0.07 | 58.96 | 0.25 | 53.51 | 0.17 |
| GraphRAG (Edge et al. (2024)) | 31.67 | 0.00 | 27.60 | 0.00 | 40.71 | 0.14 | 54.29 | 0.43 | 40.82 | 0.18 |
| HIPPORAG2 (Gutiérrez et al.) | 35.20 | 0.00 | 29.84 | 0.00 | 25.12 | 0.03 | 32.80 | 0.01 | 31.76 | 0.01 |
| RAPTOR (Sarthi et al. (2024)) | 69.88 | 0.37 | 58.92 | 0.22 | 61.36 | 0.32 | 45.62 | 0.29 | 60.69 | 0.30 |
| StructRAG (Li et al. (2025b)) | 63.68 | 0.08 | 75.25 | 0.40 | 67.57 | 0.26 | **81.75** | **0.54** | 72.62 | **0.34** |
| SchemaRAG (Ours) | 66.88 | 0.15 | **77.88** | **0.44** | **76.33** | **0.40** | 70.65 | 0.25 | **73.21** | 0.32 |
| **Set 2 (50K-100K Tokens)** | | | | | | | | | | |
| Long-context (Team (2024)) | 64.53 | 0.43 | 42.60 | 0.21 | 38.52 | 0.05 | 51.18 | 0.20 | 45.71 | 0.17 |
| Naive RAG (Lewis et al. (2020)) | 66.27 | **0.46** | 46.28 | 0.31 | 38.95 | 0.05 | 46.15 | 0.22 | 45.42 | 0.19 |
| RQ-RAG (Chan et al. (2024)) | 57.35 | 0.35 | 50.83 | 0.16 | 42.85 | 0.03 | 47.60 | 0.10 | 47.09 | 0.10 |
| GraphRAG (Edge et al. (2024)) | 24.80 | 0.00 | 14.29 | 0.00 | 37.86 | 0.00 | 46.25 | 0.12 | 33.06 | 0.03 |
| HIPPORAG2 (Gutiérrez et al.) | 38.30 | 0.00 | 43.22 | 0.00 | 55.17 | 0.00 | 24.40 | 0.00 | 37.19 | 0.00 |
| RAPTOR (Sarthi et al. (2024)) | 67.39 | 0.09 | 59.67 | 0.17 | 53.97 | 0.11 | 40.02 | 0.09 | 57.76 | 0.13 |
| StructRAG (Li et al. (2025b)) | 63.00 | 0.13 | 66.88 | 0.30 | 62.10 | **0.20** | **73.53** | **0.43** | 66.01 | **0.27** |
| SchemaRAG (Ours) | **73.62** | 0.22 | **71.30** | **0.36** | **66.33** | 0.18 | 69.38 | 0.28 | **69.30** | 0.25 |
| **Set 3 (100K-200K Tokens)** | | | | | | | | | | |
| Long-context (Team (2024)) | 46.99 | 0.27 | 37.06 | 0.13 | 31.50 | 0.02 | 35.01 | 0.07 | 35.94 | 0.09 |
| Naive RAG (Lewis et al. (2020)) | 73.69 | **0.55** | 42.20 | 0.27 | 32.78 | 0.02 | 37.65 | 0.13 | 42.60 | 0.18 |
| RQ-RAG (Chan et al. (2024)) | 50.50 | 0.13 | 44.62 | 0.00 | 36.98 | 0.00 | 36.79 | 0.07 | 40.93 | 0.05 |
| GraphRAG (Edge et al. (2024)) | 15.83 | 0.00 | 27.40 | 0.00 | 42.50 | 0.00 | 43.33 | 0.17 | 33.28 | 0.04 |
| HIPPORAG2 (Gutiérrez et al.) | 58.44 | 0.14 | 45.04 | 0.12 | 39.70 | 0.06 | 28.12 | 0.06 | 43.93 | 0.09 |
| RAPTOR (Sarthi et al. (2024)) | 65.87 | 0.19 | 58.92 | 0.32 | 50.28 | 0.18 | 34.34 | 0.09 | 51.75 | 0.15 |
| StructRAG (Li et al. (2025b)) | 67.63 | 0.24 | 68.00 | 0.27 | 63.38 | **0.22** | **68.60** | **0.41** | 66.22 | 0.26 |
| SchemaRAG (Ours) | **80.00** | 0.28 | **75.29** | **0.44** | **67.06** | 0.19 | 65.00 | 0.20 | **71.10** | **0.27** |
| **Set 4 (200K-250K Tokens)** | | | | | | | | | | |
| Long-context (Team (2024)) | 33.18 | 0.16 | 26.59 | 0.08 | 29.84 | **0.01** | 25.81 | 0.04 | 28.92 | 0.06 |
| Naive RAG (Lewis et al. (2020)) | 52.17 | **0.24** | 24.60 | 0.10 | 26.78 | 0.00 | 17.79 | 0.00 | 29.29 | 0.07 |
| RQ-RAG (Chan et al. (2024)) | 29.17 | 0.08 | 40.36 | 0.00 | 26.92 | 0.00 | 34.69 | 0.00 | 31.91 | 0.01 |
| GraphRAG (Edge et al. (2024)) | 17.50 | 0.00 | 26.67 | 0.00 | 20.91 | 0.00 | 33.67 | 0.03 | 23.47 | 0.05 |
| HIPPORAG2 (Gutiérrez et al.) | 38.15 | 0.04 | 29.11 | 0.02 | 35.77 | 0.00 | 19.25 | 0.00 | 31.87 | 0.01 |
| RAPTOR (Sarthi et al. (2024)) | 59.60 | 0.19 | 48.42 | 0.17 | 43.98 | 0.08 | 30.29 | 0.03 | 46.98 | 0.11 |
| StructRAG (Li et al. (2025b)) | **66.00** | 0.22 | 53.88 | 0.17 | 54.26 | 0.05 | **68.30** | **0.28** | 59.82 | **0.16** |
| SchemaRAG (Ours) | 64.23 | 0.13 | **65.56** | **0.30** | **60.45** | 0.07 | 63.00 | 0.17 | **63.04** | 0.15 |
| **Overall** | | | | | | | | | | |
| Long-context (Team (2024)) | 58.17 | **0.36** | 42.38 | 0.20 | 36.71 | 0.04 | 47.76 | 0.18 | 43.29 | 0.15 |
| StructRAG (Li et al. (2025b)) | 65.16 | 0.17 | 67.17 | 0.29 | 62.37 | 0.20 | **73.47** | **0.43** | 66.54 | **0.27** |
| SchemaRAG (Ours) | **72.00** | 0.20 | **73.36** | **0.40** | **68.03** | 0.22 | 68.15 | 0.24 | **69.93** | 0.26 |

**Obs1: SchemaRAG achieves state-of-the-art performance across most settings and tasks.**
SchemaRAG consistently outperforms most baselines across both LLM Score and EM rating, achieving an average LLM score of 69.93 and an EM rate of 0.26, compared to StructRAG's LLM score of 66.54 and EM rate of 0.27. These results demonstrate SchemaRAG's robust ability to handle diverse tasks and document lengths effectively. Notably, SchemaRAG underperforms compared to the long-text method and RQ-RAG on the Spot. task in Set 1. We analyze that this is due to the task understanding difficulty under this set and the low document sequence length, simpler fine-tuning-based RQ-RAG or direct long-text generation perform well without the additional design and overhead introduced by advanced RAG. These results still demonstrate SchemaRAG's superior ability to organize and utilize knowledge in complex knowledge-intensive task scenarios.

**Obs2: SchemaRAG exhibits enhanced robustness and scalability as document length and information dispersion increase.** The results across document sets reveal that baseline methods—particularly Long-context processing and Naive RAG—experience substantial performance declines of 44.75%~67.01% in both LLM Score and EM as the input context length increases. In contrast, SchemaRAG demonstrates remarkable robustness, maintaining strong performance even on the most challenging dataset, Set 4 , with only an 11.65% performance drop. Furthermore, SchemaRAG achieves the highest scores in both the Comparison task (LLM: 65.56, EM: 0.30) and the Clustering task (LLM: 60.45, EM: 0.07), while also retaining the best overall performance with an LLM Score of 63.04 and an EM of 0.15. This superior performance can be attributed to the corpus structure designed around adaptive schemas, which play a critical role in information positioning and dependency establishment for longer texts. By effectively mitigating the interference of task-irrelevant information and enhancing the capture of task-relevant content, SchemaRAG demonstrates its ability to address the challenges posed by large and complex input contexts.

**Obs3: SchemaRAG significantly advances knowledge-intensive reasoning, particularly for complex comparison and clustering tasks.** SchemaRAG's improvements are most pronounced on tasks requiring comparison and clustering. SchemaRAG outperforms StructRAG by 4.5% ~ 11.2% in overall LLM Score and EM rating. As context length increases, this advantage becomes more evident: in Set 4, SchemaRAG exceeds StructRAG by 11.4%~21.7% in LLM Score (65.56 v.s. 53.88) and by 0.25 in EM (0.30 v.s. 0.17), highlighting its superior capacity to synthesize and leverage structured knowledge for complex inferential tasks. Notably, on the chain-reasoning

Table 3: Ablation study results on the Loong benchmark.

| Method | Set 1 | | Set 2 | | Set 3 | | Set 4 | | Overall | |
|---|---|---|---|---|---|---|---|---|---|---|
| | LLM Score | EM | LLM Score | EM | LLM Score | EM | LLM Score | EM | LLM Score | EM |
| SchemaRAG | **73.21** | **0.32** | **69.30** | **0.25** | **71.10** | **0.27** | **63.04** | **0.15** | **69.93** | **0.26** |
| w/o ST | 67.34 | 0.27 | 64.81 | 0.19 | 64.45 | 0.18 | 59.68 | 0.07 | 64.18 | 0.18 |
| w/o SC | 63.24 | 0.29 | 62.64 | 0.17 | 61.75 | 0.17 | 58.34 | 0.09 | 61.95 | 0.18 |
| w/o SU | 69.42 | 0.35 | 66.82 | 0.23 | 67.96 | 0.24 | 61.98 | 0.09 | 66.30 | 0.24 |

Figure 3: The case study of comparison between our SchemaRAG and the baseline StructRAG.

task, SchemaRAG performs slightly worse than StructRAG, we analyze this due to the task's inherent complexity and reliance on precise multi-step reasoning. StructRAG's fixed templates, tailored specifically for such reasoning tasks, appear to provide an advantage over SchemaRAG's adaptive strategies in this scenario. Notably, on the chain-of-reasoning task, SchemaRAG performs slightly worse than StructRAG. We attribute this to the task's inherent complexity and its reliance on precise multi-step reasoning. The fixed templates of StructRAG, specifically designed for such reasoning tasks, seem to offer an advantage over SchemaRAG's adaptive strategies in this context.

### 4.3 CASE STUDY (RQ2)

**Obs4: SchemaRAG's adaptability enables accurate citation relationship identification, outperforming rigid template-based methods like StructRAG.** We present a case study in Figure 3 on constructing citation chains across scholarly documents. StructRAG, reliant on fixed templates, fails to extract citation relationships, misidentifying mutual citations between papers A and B due to incorrect focus on statements in the introduction and formulas in the method section. SchemaRAG addresses this with two key improvements: flexible structural descriptions and strategy guidance. It emphasizes focusing on paper titles and reference lists while providing clear steps for constructing schemas with causal chains and chunk types, significantly improving accuracy and contextual alignment. More detailed analysis of intermediate results based on this case and various other scenarios, as outlined in the Appendix C.4.

### 4.4 ABLATION ANALYSIS (RQ3)

**Obs5: Adaptive schemas are crucial for knowledge-intensive reasoning, with schema construction contributing most significantly to model performance.** We conducted ablation studies (Table 3) to evaluate the contributions of schema thinking (w/o ST), schema construction (w/o SC), and schema utilization (w/o SU) in SchemaRAG. Removing any component significantly reduces performance, highlighting their interdependence. Notably, the w/o ST configuration removes the adaptive strategy generation mechanism, while w/o SC signifies that all documents are processed

Table 4: Cost Comparison Across Construction, Utilization, and Total Phases on the Loong Benchmark with Baseline Models.

| Phase | Construction Phase | | | Utilization Phase | | | Total Phase | | |
|---|---|---|---|---|---|---|---|---|---|
| Model | Tokens | API Calls | Running Time | Tokens | API Calls | Running Time | Tokens | API Calls | Running Time |
| Long-Context | - | - | - | 108K | 9.85 | 1.3 min | 108K | 9.85 | 1.3 min |
| RQ-RAG | - | - | - | 126K | 10.25 | 1.2 min | 126K | 10.25 | 1.2 min |
| GraphRAG | 250K | 25.00 | 215.3 min | 45K | 6.50 | 1.8 min | 295K | 31.5 | 217.1 min |
| StructRAG | 86K | 9.85 | 8.2 min | 37K | 4.00 | 1.5 min | 130K | 13.85 | 9.7 min |
| SchemaRAG(Ours) | 87K | 8.95 | 8.6 min | 19K | 10.85 | 1.6 min | 162K | 22.70 | 10.36 min |

as a flat, long context without structural organization. Meanwhile, w/o SU eliminates both task structuring and the hierarchical retrieval-merge strategy, leading to direct answer generation solely based on the knowledge schema. It is evident that the performance degradation associated with w/o SC is the most pronounced, reaching 11.41%. Secondly, the w/o ST configuration also results in a notable decrease of 8.22%. These findings underscore the critical importance of adaptive schemas in facilitating knowledge-intensive reasoning tasks.

## 4.5 RESOURCE COSTS ANALYSIS (RQ4)

We assess SchemaRAG's cost-efficiency on Loong, reporting average runtime, token usage, and API calls per query across Construction, Utilization, and Total phases. Unlike GraphRAG that prebuild a corpus-wide offline index, we adapt GraphRAG to construct the corpus per query—following StructRAG's statistical protocol—to ensure a fair comparison aligned with our setup.

**Obs6: SchemaRAG offers a strong cost–performance balance with reasonable resource use, despite added overhead for complex tasks** The comparison results are presented in Table 4, based on experiments conducted with an A100 GPU and the Qwen-2-72B-Instruct model. In the construction phase, GraphRAG incurs the highest costs, while StructRAG and SchemaRAG achieve significantly lower runtimes of 8.2 and 8.6 minutes, respectively. In the Utilization phase, SchemaRAG uses refined knowledge schemas as input, significantly reducing token consumption. However, its hierarchical retrieval-merge mechanism, involving query tree traversal, justifies a higher number of API calls for more advanced processing and effective resolution of complex tasks. In the Total phase, SchemaRAG's token consumption and API calls are marginally higher than those of StructRAG, but still far more efficient than GraphRAG. Further details appear in Table 6, Appendix C.4.

## 5 RELATED WORK

RAG ( Gao et al. (2023); Wu et al. (2024b); Li et al. (2022); Chen et al. (2024a); Lan & Jiang (2021)) improves LLMs by incorporating external documents, but it struggles to capture deep, global associations in fragmented domains, limiting comprehensive knowledge integration. Recent studies( Han et al. (2024); Zhang et al. (2025); Jiang et al. (2025)) focus on integrating explicit structural representations into RAG systems' retrieval and reasoning pipelines. GraphRAG approaches ( Edge et al. (2024); He et al. (2024); Guo et al. (2024)) have explored constructing graph communities or graph-based databases, as well as transforming retrieved passages or entire corpora into structured formats such as knowledge graphs or triplet-based fact representations. Some studies ( Luo et al. (2023); Zhao et al. (2024); Li et al. (2025a)) leverage knowledge graphs to link document or knowledge entities and to identify effective reasoning paths for multi-hop question answering. While effective, these methods often rely on single-type structures or fixed templates, limiting flexibility and adaptability. Our approach introduces adaptive schemas that dynamically organize knowledge to meet task-specific requirements, enhancing flexibility and generation.

## 6 CONCLUSION

In this paper, we propose SchemaRAG, an adaptive schema-guided RAG framework inspired by schema theory in cognitive science. SchemaRAG dynamically constructs and utilizes knowledge schemas tailored to each query and its associated documents, enabling more effective integration and reasoning over heterogeneous information sources. Extensive experiments on real-world benchmarks demonstrate that SchemaRAG can flexibly organize, extract, and synthesize relevant information for complex tasks, significantly improving performance over existing approaches.

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

## A    ETHICS STATEMENT

This study complies with the ICLR Code of Ethics. It involved no human or animal subjects. All datasets, including Loong dataset, were sourced responsibly, adhering to relevant usage guidelines and respecting privacy standards. We have been diligent in mitigating potential biases or discriminatory impacts throughout our research, ensuring no personally identifiable data was utilized or any experiments conducted that might compromise privacy or security. Our commitment is to uphold transparency and integrity across all stages of the research.

## B    REPRODUCIBILITY STATEMENT

We have taken extensive steps to ensure the reproducibility of the results presented in this paper. All code and datasets will be made available in a GitHub repository upon the paper's acceptance, facilitating easy access to encourage replication and verification. The experimental procedures, including detailed descriptions of the training stages, model setups, and hardware specifications, are thoroughly documented in the paper. Furthermore, a comprehensive outline of SchemaRAG is provided to support the recreation of our experiments.

Furthermore, the Loong dataset ensure that consistent and comparable evaluation results can be obtained across different setups. We believe these efforts will enable fellow researchers to replicate our work, thereby contributing to advancements in the field.

## C APPENDIX

### C.1 LLM USAGE

In preparing this manuscript, Large Language Models (LLMs) were employed to support the writing and editing process. The LLM assisted in refining the manuscript by enhancing the language, improving readability, and ensuring clarity. Tasks included sentence restructuring, grammar correction, and optimizing the overall text flow. It's crucial to emphasize that the LLM did not partake in ideation, research methods, or experimental design. These components, including all conceptual and analytical work, were exclusively the authors' contributions. The involvement of the LLM was limited to linguistic enhancements without affecting the scientific content or data analysis.

### C.2 LIMITATIONS

This study presents several key limitations that warrant consideration. Firstly, our method involves additional inferences, which leads to increased token resource consumption. Secondly, our experiments primarily focus on domain-specific data, without extending the analysis to open-domain question answering. Thirdly, the evaluation metrics in this study primarily depend on LLM-based assessments, as there are no established benchmarks for evaluating the quality of structured results. This reliance may impact the robustness and comparability of our evaluation outcomes. To address this, we conducted human evaluations of the generated content and intermediate results, providing correlation testing with the automated scores from LLMs. Lastly, while our research emphasizes the generative aspect of RAG technology, it does not extensively address the retrieval component. This could limit the comprehensiveness of our approach in fully leveraging RAG capabilities. We acknowledge these limitations and plan to address them in future work, aiming to enhance the efficiency, generalizability, and evaluation rigor of our method.

### C.3 RELATED WORK

**RAG.** RAG ( Gao et al. (2023); Wu et al. (2024b); Li et al. (2022); Chen et al. (2024a); Es et al. (2024); Lan & Jiang (2021); Lyu et al. (2025)) enhances LLMs by integrating external documents, boosting factuality and reducing hallucinations in knowledge-intensive tasks ( Lewis et al. (2020)). Early RAG ( Guu et al. (2020); Lewis et al. (2020)) employed single-step retrieval with fixed passages, limiting multi-step reasoning. Recent advances ( Qi et al. (2019); Shao et al. (2023)) enable adaptive and iterative retrieval, where models dynamically trigger searches or decompose queries based on uncertainty or reasoning needs. Later methods ( Shao et al. (2023); Guan et al. (2024); Qi et al. (2019)) iteratively refine evidence across multiple cycles to improve multi-hop question answering. Despite their effectiveness, these approaches still struggle to capture global associations in domains with fragmented knowledge, limiting comprehensive task-relevant integration.

**Structure-Enhanced RAG Methods.** Recent research ( Han et al. (2024); Zhang et al. (2025); Jiang et al. (2025)) has increasingly focused on integrating explicit structural representations into the retrieval and reasoning pipeline of RAG systems. Early GraphRAG approaches ( Edge et al. (2024); He et al. (2024)) have explored constructing graph communities or graph-based databases ( Guo et al. (2024)), as well as transforming retrieved passages or entire corpora into structured formats such as knowledge graphs or triplet-based fact representations ( Jimenez Gutierrez et al. (2024); Zhao et al. (2024)). Some studies ( Panda et al. (2024); Zhao et al. (2024)) leverage knowledge graphs to link document or knowledge entities and to identify effective reasoning paths for multi-hop question answering. Representative methods, such as RoG ( Luo et al. (2023)) and KnowTrace ( Li et al. (2025a)), dynamically construct relevant knowledge graphs or plan inference paths by autonomously tracking the required knowledge triplets. Additionally, several works ( Edge et al. (2024); Sarthi et al. (2024); Jimenez Gutierrez et al. (2024)) propose structured indexing or retrieval mechanisms to enhance the reliability of domain-specific responses. For instance, MedGraphRAG ( Wu et al. (2024a)) constructs multi-level citation graphs and designs structured retrieval mechanisms to ensure trustworthy medical responses, while DeepSolution ( Li et al. (2025c)) addresses complex, constraint-driven engineering problems through tree-structured exploration and reasoning mechanisms. Despite their advantages, these methods still typically rely on single-type structure or fixed pre-defined construction templates, which limits their flexibility and task adaptability. In contrast, our approach introduces adaptively constructed schemas that dynamically organize and utilize

Table 5: Comparison of LLM Scores and Human Evaluations Across Domains and Structure Types.

| Domain | Structure Types | LLM Score | Human Evaluation | | | |
|--------|-----------------|-----------|------------|----------------|--------------|------------|
| | | | For Answer | For TaskParser | For Strategy | For Schema |
| **Finance** | Catalogue Group ‖ Table | 77.5 | 81.0 | 96.0 | 88.0 | 92.0 |
| **Legal** | Catalogue Group ‖ Chunk | 76.5 | 72.0 | 84.0 | 78.0 | 84.0 |
| **Academic** | Correlation Graph ‖ Chunk | 62.5 | 55.0 | 75.0 | 66.0 | 45.0 |
| **Correlation Coefficient with LLM Score** | | | 0.993 | 0.990 | 0.988 | 0.851 |

Table 6: Resource Consumption Analysis for Specific Task Cases in the Loong Dataset.

| CaseID | Task Type | Domain | Chosen Structure Types | Decomposed Query Number | Documents Number | Tokens Number of Docs | Tokens Number of Consumed | Total Number of API Calls | Running Time |
|--------|-----------|--------|------------------------|-------------------------|------------------|-----------------------|---------------------------|---------------------------|--------------|
| Case 1 | Comp. | Financial | Group-Table | 1 | 6 | 82449 | 105542 | 12 | 2.74 min |
| Case 2 | Spot. | Financial | Group-Table | 1 | 6 | 160413 | 189333 | 7 | 6.34 min |
| Case 3 | Clus. | Financial | Group-Table | 3 | 6 | 136907 | 167011 | 12 | 4.65 min |
| Case 4 | Clus. | Academic | Graph-Chunk | 2 | 3 | 45132 | 67942 | 7 | 5.2 min |
| Case 5 | Clus. | Academic | Graph-Chunk | 2 | 11 | 206722 | 162832 | 17 | 25.43 min |
| Case 6 | Chain. | Legal | Chunk-Chunk | 4 | 4 | 19374 | 37493 | 11 | 2.87 min |
| Case 7 | Chain. | Legal | Group-Chunk | 23 | 23 | 86994 | 474931 | 50 | 32.7 min |
| Case 8 | Chain. | Legal | Group-Chunk | 4 | 43 | 103552 | 195815 | 51 | 12.52 min |
| Case 9 | Chain. | Academic | Chain-Chunk | 4 | 4 | 149024 | 176552 | 11 | 13.36 min |

knowledge structures with task-specific requirements, addressing the rigidity of prior methods and improving generation.

## C.4 EXPERIMENTS

**Correlation Analysis Between Human Evaluation and LLM-Judged Scores.** The quality assessment benchmarks employed in this study are primarily based on automated scoring by LLMs, consistent with the baseline settings in prior work Li et al. (2025b). To mitigate potential controversies and instability associated with automated evaluations, we designed a correlation test to compare human evaluations with LLM-based evaluations. We randomly selected 20 task samples from each of the three domain attributes in the Loong dataset. Additionally, three senior doctoral students were invited to serve as human experts and perform a manual evaluation of the SchemaRAG processing results for these 20 samples. The evaluation criteria extended beyond the quality of the final answer generation and also encompassed intermediate outcomes, including task parsing results, strategy generation results, and schema construction results produced during the SchemaRAG workflow. For each domain, we computed the average LLM score across the 20 samples as a reference value, alongside the average scores provided by the three human experts for the same data. As presented in Table 5, all correlation coefficients for the indicators exceed 0.8, indicating a very strong and significant correlation between the LLM scores and the human evaluations. Furthermore, the human experts' assessments of the intermediate results fall within an acceptable and reasonable range. These findings suggest that the automated scoring provided by large language models serves as a reliable indicator of the generation quality in SchemaRAG.

**Resource Consumption Analysis.** Although Table 4 outlines the average resource consumption per query, Table 6 provides a more detailed and intuitive analysis of the resource overhead associated with processing individual cases in the Loong dataset using SchemaRAG. It presents statistics for nine representative cases, including task attributes (task type and domain), the structure types chosen by SchemaRAG, the inherent workload of the input (query complexity, number of documents, and total tokens in the documents), and the resource consumption metrics observed during execution (total tokens consumed, total API calls, and runtime). The number of queries decomposed through task structuring in SchemaRAG serves as an approximate indicator of the task's complexity and difficulty. We can observe that the additional token consumption by SchemaRAG, compared to the original document tokens, is influenced by both task complexity and the number of documents. Moreover, as the number of input documents increases and the query becomes more complex, the number of API calls and runtime of SchemaRAG rise significantly. For instance, in cases such as Case 1, Case 8, and Case 9, although the number of tokens contained in the documents is similar, there are notable differences in resource consumption.

**Statistical Analysis of Intermediate Results.** To provide a clearer understanding of the index structure selection and knowledge structure selection in SchemaRAG, we conducted a statistical

Table 7: Win rate of SchemaRAG vs. baselines on Podcast Transcripts.

| Compared Method Pair | Comprehensiveness | Diversity | Empowerment | Directness | Average |
|---|---|---|---|---|---|
| SchemaRAG vs. Long-context | 99 | 94 | 98 | 99 | 97.50 |
| SchemaRAG vs. RAG | 74 | 85 | 66 | 75 | 75.00 |
| SchemaRAG vs. RAPTOR | 70 | 85 | 50 | 66 | 67.75 |
| SchemaRAG vs. GraphRAG | 68 | 77 | 42 | 55 | 60.50 |
| SchemaRAG vs. StructRAG | 63 | 58 | 46 | 55 | 55.50 |

analysis of the structural types used in schemas constructed from 1,600 samples in the Loong dataset, as illustrated in Figure 4. The analysis reveals that catalogue group dominate the index structures, while the knowledge structures are primarily composed of tables and chunks. This distribution can be attributed to the characteristics of the Loong dataset, which consists predominantly of financial reports and legal documents and involves a significant number of matching and clustering tasks. To further validate the rationale behind the structure selection, we present visualizations of the two most frequently used schema types, demonstrating their alignment with the dataset's task requirements.

**Win-Rate Comparison Against Baselines.** We conduct a set of win-rate experiments on SchemaRAG vs. baselines on the Podcast Transcripts dataset. The comparison was conducted across four dimensions, following the verification methods used in GraphRAG and StructRAG. The results are as Table **??**. The results show that SchemaRAG achieves the best average performance in Comprehensiveness, Diversity, Empowerment, and Directness in podcast domain.

**Case Studies of Intermediate Results.** In this section, we present a comprehensive visualization of SchemaRAG's end-to-end processing workflow, starting with a case study and illustrating the intermediate outputs at each step. Figures 5 and Figures 6 showcase two representative case studies: one focused on a financial task and the other on an academic task.

In the financial task depicted in Figure 5, SchemaRAG demonstrates a systematic and logical derivation process encompassing task structuring, strategy generation, schema construction, and schema utilization. This process ultimately results in the accurate clustering of three companies, highlighting the model's capability to handle complex organizational tasks effectively.

Figure 6 delves into the academic task, which builds upon the example introduced in Figure 3. Here, a more detailed analysis of the intermediate steps is provided. For the challenging task of constructing academic paper citation chains, SchemaRAG successfully generates a multi-level query tree and organizes the citation relationships within the schema index, effectively capturing the intricate hierarchical structure of academic references.

These case studies highlight that SchemaRAG's effectiveness lies in its ability to produce a highly structured and concise representation of both task requirements and document content. Its adaptive guidance strategy, which seamlessly integrates schema construction with content generation, is driven by a nuanced understanding of the task at hand.

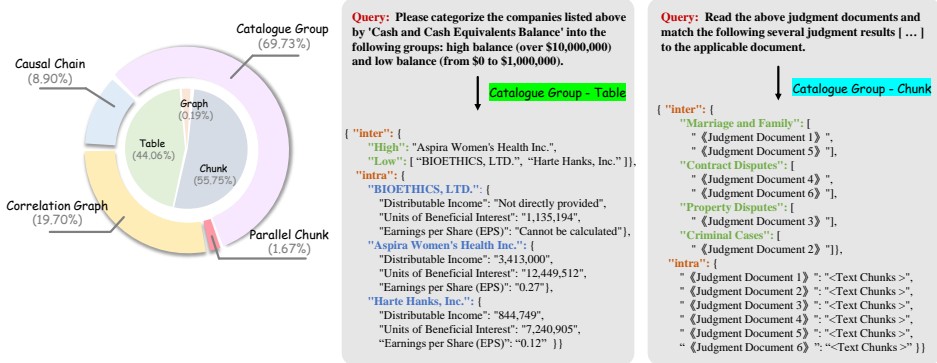

Figure 4: Distribution of Structure Types and Examples of Schema Representations.

Figure 5: The case study of a financial task utilizing SchemaRAG.

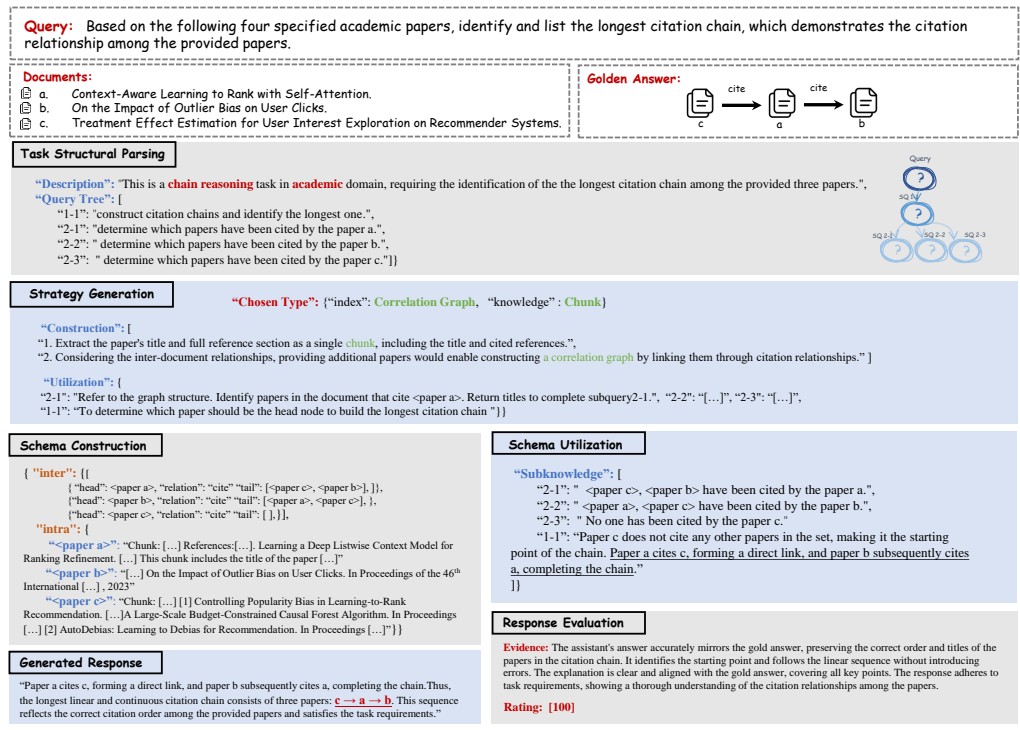

Figure 6: The case study of a academic task utilizing SchemaRAG.

# D PROMPTS DESIGN AND SOME CASES

We provide key system prompts related to the pipeline to help clarify our workflow. Due to space constraints, we omit some output-format specifications and a subset few-shot examples from the prompt templates. Content enclosed in curly braces denotes the original input or an intermediate input for the task.

We also release more illustrative cases via an anonymous GitHub link: `https://anonymous.4open.science/r/SchemaRAG_cases-58F4/`. These representative datasets include end-to-end examples from diverse domains, task types, and text lengths, covering the input and output at each stage, the constructed schema, the final answer, and the LLM-assigned score.

---

**Task Structural Parsing**

```
#Instruction

In order to solve complex document-related problems, you need
to break down the given Query into multiple relatively simple
and independent sub-queries with a tree structure. Perform the
step-by-step analysis, obtain a query tree and finally follow
the Output Requirements and Output Template to return the final
result.

Step 1: Task Understanding
Please provide a brief analysis and explanation of the core
requirements of this task. This should not exceed 100 words
and should be based on the query and the document information
provided.

Step 2: Task Type
Consider how to extract information from the document and how
to process the relationships between different entities in
the query. From the following list [Information Extraction,
Matching, Comparison, Clustering, Chain Reasoning, Open-Ended
Thinking], select one term that best represents the type of
task for this query.

Step 3: Question Decomposition
Break down the given query into sub-questions and construct a
logical tree structure. The query acts as the parent node,
and you should rewrite or further decompose it into easily
understandable sub-questions as child nodes.

#Output Template:
{output_temp}

#Examples:
{example_output}

#Doc Info:
{titles}

#Query:
{query}

#Output:
```

**Strategy Generation**

**#Instruction**

This is a {task_type} task, defined as {task_description}.
Given the query list {queries} and multiple document similar
to the example document, you need to analyze and locate the
distribution of important information.
Please think through the following steps and generate some
executable program steps as a strategy to guide the structured
organization of document information and the retrieval of
task-relevant information, and ensure your answers adhere to
the specified output requirements.

**Step 1:** Analyze the task requirements and identify where
information relevant to each query in the query list is
distributed within the documents. [...]

**Step 2:** Based on the analysis above, select the most
suitable structure types for both inter-document relations and
intra-document organization, tailored to the task requirements.
- For inter-document relations, select one structure type from
[parallel chunks / catalogue group / causal chain / correlation
graph], and output as <Inter_chosen>.
- For intra-document organization, select one structure type
from [graph / table / chunk], and output as <Intra_chosen>.

**Step 3:** Output <Construction_Planning> as Instructions for
Structuring Document Content.
Using the results from Step 2, as well as the given task
description and example document content, generate a detail
ed construction plan for document structuring. The plan should
specify:
- "intra" key: Clearly describe how to organize knowledge
within each document, as the type <Intra_chosen>. List
the precise steps to follow (such as extraction, matching,
summarization, calculation, or chain-of-thought reasoning).
Explicitly specify which key information should be retained
from each document.
- "inter" key: Clearly describe how to establish associations
or relationships between multiple documents, as the type
<Inter_chosen>. List the exact and actionable steps required
to build these inter-document relationships.[...]

**Step 4:** Output <Retrieval_Planning> as Instructions for
Retrieving Task-relevant Knowledge.
Please use the structured information obtained by Step 3 as
the sole knowledge source for retrieval. For each subquery
node in the query tree, clearly specify an executable retrieval
plan|detailing step-by-step operations|to extract task-relevant
knowledge from this source.[...]

#Example: {example_output}
#Output Template: {out_temp}
#Example Document Content: {content}
**#Output:**

**Knowledge Structure Construction**

```
#Instruction

This is a {task_type} task, defined as {task_description}.
As an assistant, reorganize the provided Document Content
into structure information based on the specified
structure type ({intra_type}). Refer to the Construction
Requirements({intra_requirements}) and follow the
implementation steps outlined in the Construction Plan.

Notes:
- Language Requirement: When extracting and reorganizing
content, always preserve the document's original language. Do
not translate.
- Output Requirement: The final output must consist solely
of knowledge text in the consistent format (chunks, table,
or triples) as specified. No additional explanatory content
should be included.

#Construction Plan:
{planning}

#Document Content:
{content}

#Output:
```

**Schema Construction**

```
#Instruction

The task description is {task_description} and task type
is {task_type}. Construct nested structured knowledge in
JSON format using the provided intra-document content.
Specifically, treat each piece of information within the
provided intra-document content as independent knowledge units.
Then, use the relationships between documents ({inter_type})
to construct the external structure of multiple knowledge
units, following the Requirements: ({inter_requirements}).
The specific operational method refers to Construction Plan.

- The value of "inter" captures knowledge about inter-document
relationships.
- The value of "intra" is a dictionary whose keys are the names
of each document, and whose values represent intra-document
knowledge for the corresponding document.

#Construction Plan:
{planning}

#Output Template:
{temp_structure}

#Intra-document Content:
{intra_knowledge}

#Output:
```

**Knowledge Retrieval**

**#Instruction**

According to the Query, filter the Knowledge Content to retain only the relevant information that can help answer the Query. Refer to the Retrieval Plan planning and carefully analyze the attributes and entities mentioned in the Query.

Note:
- Knowledge content:  A JSON-formatted object with three keys:  details (the value denotes key content extracted from individual documents), inter (the value denotes relationships among multiple independent documents/entities) and intra (the value denotes the summarized content extracted from each document)
- Language Requirement:  When analyzing queries and extracting content, always preserve the original language, do not translate.

#Output Template:
{out_temp}

#Query:
{subquery}

#Knowledge content:
{schema}

**#Output:**

