# OpenReview forum: "SchemaRAG: Enhancing Knowledge-Intensive Reasoning of LLMs via Inference-Time Adaptive Schema"
_ICLR.cc/2026/Conference — Submitted to ICLR 2026_

### Official Review · Reviewer_mGUh · 2025-10-30

**Soundness:** 2
**Presentation:** 2
**Contribution:** 2
**Rating:** 4
**Confidence:** 3

**Summary:**

The paper proposes SchemaRAG, an adaptive, schema-guided framework to improve multi-document reasoning in Retrieval-Augmented Generation. Unlike prior approaches that rely on rigid structural templates (e.g., fixed graphs, tables, or chunking), SchemaRAG dynamically organizes factual evidence according to the query’s needs. Given a query and retrieved documents, the system decomposes the query into sub-problems, devises strategies for constructing a fit-for-purpose schema, and then uses that organized knowledge to generate the final answer.

**Strengths:**

1. Better multi-document integration: Organizes evidence to capture critical dependencies that static templates often miss, improving cross-document reasoning.

2. Interpretability: The constructed schema provides an explicit intermediate representation.

3. Empirical strength: Demonstrates gains over strong baselines.

**Weaknesses:**

1. The advantage over StructRAG is unclear. The paper doesn’t clearly isolate what SchemaRAG adds beyond StructRAG’s sub-questioning and format selection. Table 1’s “composite structures / construction / utilization” aren’t rigorously defined. Why and how each of “composite structures,” “construction,” and “utilization” impacts performance. The case study in Figure 3 seems driven by StructRAG’s graphing error, not by a principled limitation.

2. Many steps rely on LLM prompts, but prompts (and prompt engineering choices) aren’t provided (even in appendix). The prompts may help to make the method clearer.

3. Whether multiple full documents are first retrieved and then reorganized for utilization. If so, it seems bring large overhead compared to the ones retrieve a piece of knowledge. Table 4 shows no significant extra cost, especially in the Construction phase. Could you provide more explanation?

**Questions:**

Sea the weaknesses.

---

> ### Author Response · Authors · 2025-11-25
> **Rebuttal by Authors (1/2)**
>
> We appreciate your efforts of our work. To address your concerns, we provide point-by-point responses. We have summarized your review comments as [Q1] to [Q5].
>
>
> **[[Q1]]: The advantage over StructRAG, with additional explanation.**
> > **Comment:**  The advantage over StructRAG is unclear. The paper doesn’t clearly isolate what SchemaRAG adds beyond StructRAG’s sub-questioning and format selection. The case study in Figure 3 seems driven by StructRAG’s graphing error, not by a principled limitation.
>
> - **StructRAG’s Limitation:** StructRAG accurately identified a graph as the appropriate structure for academic paper citation tasks. When further constructing citation relationships, each academic paper was combined with a fixed graph construction template as input and fed into the LLM for inference. The relationship capture result for papers a, b, and c is shown in the "constructed graph" section.
>
> Upon examining the intermediate inference process, we found that the error occurred because the LLM did not know where to focus in the paper to solve the current task; some citation relationships were captured based on incorrect statements in the introduction and formulas in the method section. However, due to the lengthy inference content and paper titles, we only present the key erroneous triples in Figure 3.
>
> - **SchemaRAG’s Advantages:** SchemaRAG offers two key improvements over StructRAG: 1) More flexible structural description, 2) Strategy guidance.
> As shown in the Figure 3, the thinking process generates a strategy based on task requirements, emphasizing focus on the paper title and reference list, and providing specific steps for constructing a schema with causal chain and chunk type. This improves the accuracy of the structure, thereby enhancing the quality of the final answer.
>
> We have also provided more detailed case studies, which can be found in Figures 5-6 of Appendix C.4.
>
> Based on your suggestion, we will highlight the key information in Figure 3 in the revised version and incorporate the above content into the case study in Sec 4.3 to further enhance the discussion.
>
>
>
> **[[Q2]]: Strict definition of some statement.**
> > **Comment:**  Table 1’s “composite structures / construction / utilization” aren’t rigorously defined. Why and how each of “composite structures,” “construction,” and “utilization” impacts performance.
>
>
> We provide the following definitions, which can be understood in conjunction with the content of Figure 1 and PRELIMINARIES.
>
> - **Composite Structures:** It can be considerd as a structure of structures, which can include multiple structure types, unlike selecting a single structure as a form of text content recombination. For example, the knowledge schema is composed of an knowledge structure nested within a index structure, which is defined in detail in Sec.2.2.
>
> - **Construction:** The process of reorganizing the original document into a structured form of factual knowledge.
>
> - **Utilization:** The process of retrieving question-related information from structured knowledge and merging it into the final answer.
>
> Based on your suggestion, we will integrate the rigorous definitions of these three concepts into the questions in PRELIMINARIES and Table 1 in the final version.
>
>
>
> **[[Q3]]: Why and how the construction and utilization of schema impacts performance.**
> > **Comment:**  Why and how each of “composite structures,” “construction,” and “utilization” impacts performance.
>
> Beyond the schema theory introduced in the introduction, we offer a more detailed discussion as follows:
>
> - **Composite structures:**  We argue that selecting multiple structures and forming a structure of structure based on scenario needs is more conducive to flexibly characterizing the hierarchy and relationships between information.
>
> - **Construction:** As per the consensus of related structured RAG work, the advantage of sampling and recombining raw document information into structured information lies in its ability to explicitly construct relationships between documents or facts, capture fragmented information, and shorten the reasoning path of dense, long text knowledge, thereby improving problem-solving capabilities.
>
> - **Utilization:** We argue that after acquiring refined content and information relationships through construction, hierarchically decomposing complex problems according to solution logic is also crucial. Just as the human learning process breaks down complex problems into several simple sub-problems, then solves them one by one and executes the final decision.
>
> Our research focuses on utilizing the idea of ​​adaptive schemas to stimulate the model's potential for better information integration from flattened text in complex task scenarios. It possesses the ability to generate task-customized strategies to guide subsequent construction and utilization processes.

---

> > ### Author Response · Authors · 2025-11-25
> > **Rebuttal by Authors (2/2)**
> >
> > **[[Q4]]: Supplement to LLM prompt.**
> > > **Comment:**  Many steps rely on LLM prompts, but prompts (and prompt engineering choices) aren’t provided (even in appendix). The prompts may help to make the method clearer.
> >
> > - We found that adaptive schema designed based on the LLM prompt have achieved good results.
> >
> > - Future work will further explore incorporating RL-guided agents for strategy generation, utilizing multi-agent workflows to further improve the framework's applicability and stability in complex scenarios.
> >
> > - We promise to release all code and system prompts immediately upon paper acceptance. Based on your suggestions, we will provide all system prompts involved in the pipeline in the appendix of the final version of the paper to assist readers in understanding.
> >
> >
> >
> > **[[Q5]]: The discussion of resource consumption.**
> > > **Comment:**  Whether multiple full documents are first retrieved and then reorganized for utilization. If so, it seems bring large overhead compared to the ones retrieve a piece of knowledge. Table 4 shows no significant extra cost, especially in the Construction phase. Could you provide more explanation?
> >
> > - Knowledge-Intensive Task Focus: It is worth noting that our research focuses on knowledge-intensive reasoning tasks. As described in Sec2.1, we use a task query and a set of task-related documents as the raw input for each inference step, and perform structured reorganization and utilization of information at inference time. **This differs from some GraphRAG methods that first construct an offline index among all documents before performing relevant corpus retrieval and inference generation.**
> >
> > - Statistical Method: To ensure comparison fairness, we adjusted GraphRAG to construct the corpus on a **query-by-query basis**, following the statistical approach and discussion of StructRAG, which is consistent with our task setup.
> >
> > We have also provided a comparison with existing Structure-Based RAG methods, claiming the innovations of our SchemaRAG compared to other works and its different task applicability.
> > | **Structure-based RAG**  | **Task**      | **Input Content**    | **Offline Indexing**       | **Inference-Time Structuring**  | **Composite Structures**   | **Adaptive Strategy Guidance** |
> > |--------|----------------|------------------|------------------|------------------|------------------|------------------|
> > | GraphRAG           |   QFS    |     Query+ All Documents       |      ✅       |        ❌     | Graph+Chunk  |  ❌
> > | StructRAG          |   Knowledge-Intensive Reasoning     |      Query+ Query-wise Documents      |      ❌                 |  ✅   | Graph/Chunk/Algorithm ... |  ❌
> > | **SchemaRAG (Ours)**         |       Knowledge-Intensive Reasoning      |   Query+ Query-wise Documents           |    ❌      |  ✅  |    Flexible Schema     |  ✅
> >
> >
> > Based on your suggestion, we will supplement the above reasons and strengthen the discussion in Sec.4.5.
> >
> >
> >
> > -----------------
> > If you have any further concerns of our work, we would be happy to address them during the comment stage.
> >
> > We look forward to your reply!
> >
> > Best regards,
> >
> > Authors.

---

> > > ### Comment · Reviewer_mGUh · 2025-11-27
> > >
> > > As you mentioned, compared with StructRAG, the proposed method claims two main advantages: (1) more flexible structural descriptions and (2) strategy guidance. However, the techniques that support these claims are not clearly described. Many key steps appear to rely on prompting the LLM, yet the actual prompts are not provided. As a result, the technical novelty of the paper remains unclear.

---

> > > > ### Author Response · Authors · 2025-12-02
> > > > **Rebuttal to Final Concerns from Reviewer mGUh**
> > > >
> > > > **Firstly**, we emphasize that our technological innovations are clearly detailed in Table 1 and Figure 1, with all technical information included in the method section of **the original paper**.
> > > >
> > > > - **Technical Details:** Schema construction, including flexible structural descriptions, is explained in Sec 3.2, while adaptive strategy guidance is detailed in Sec 3.1. Figure 2 provides an overview of the pipeline.
> > > > - **Our Contribution and Technological Innovation:** 1) We introduce adaptive schemas to address limitations of existing structured RAGs; 2) SchemaRAG can adaptively construct and utilize the most suitable knowledge schemas according to different specific tasks, significantly enhancing knowledge-intensive reasoning.
> > > >
> > > > ----------
> > > > **Secondly,** we respectfully refute the claims that "Many key steps appear to rely on prompting the LLM" and "As a result, the technical novelty of the paper remains unclear." .
> > > > We have reviewed numerous outstanding and representative existing works, as summarized in the table below.
> > > >
> > > >
> > > > | **Structure-based RAG**  | **Research Area**      | **Focus Task**    | **Key Technology Support**       | **Publication Venue / Impact**  |
> > > > |--------|----------------|------------------|------------------|------------------|
> > > > | GraphRAG      | RAG |  QFS  |  LLM Prompts + Community Detection      |   arxiv'24 (Citation 1149)     |
> > > > | ArchRAG       | RAG  |  QFS  |   LLM Prompts  + Community Detection      |   arxiv'25  (Citation 60)   |
> > > > | HippoRAG2    |  RAG |  Muti-hop Reasoning |   LLM Prompts     |     ICML'25      |
> > > > | LogicRAG [1]  |   RAG |  Muti-hop Reasoning  |   LLM Prompts  |     AAAI'26 |
> > > > |  KAR^3-RAG [2] |   RAG |   Muti-hop Reasoning      |   LLM Prompts  |     ICML'25   |
> > > > |  KnowTrace [3] |   RAG |   Muti-hop Reasoning      |   LLM Prompts  |     KDD'25   |
> > > > | StructRAG   |   RAG   |   Knowledge Intensive Reasoning   |    LLM Prompts      |  ICLR'25     |
> > > >
> > > > *[1] You Don’t Need Pre-built Graphs for RAG: Retrieval Augmented Generation with Adaptive Reasoning Structures.*
> > > >
> > > > *[2] From Complex to Atomic: Enhancing Augmented Generation via Knowledge-Aware Dual Rewriting and Reasoning.*
> > > >
> > > > *[3] KnowTrace: Bootstrapping Iterative Retrieval-Augmented Generation with Structured Knowledge Tracing.*
> > > >
> > > > **Using LLM prompts as a key technical tool does not imply a lack of innovation; instead, it enables advancements in no-training paradigms and some agent paradigms.**
> > > >
> > > > ------
> > > > **Finally,** regarding your concerns about "not clearly described" and "technical novelty remains unclear," we consider **this concern arises solely from the fact that the corresponding LLM prompts were not released.**
> > > >
> > > > To address this, we have provided the key system prompts related to formulas (3), (4), and (6), now included in **Appendix D of the revision version.**
> > > > Additionally, we have added numerous cases showcasing different schema types across various tasks, available through the [anonymous link](https://anonymous.4open.science/r/SchemaRAG_cases-58F4/).

---

### Official Review · Reviewer_Urh7 · 2025-11-01

**Soundness:** 2
**Presentation:** 2
**Contribution:** 2
**Rating:** 4
**Confidence:** 4

**Summary:**

This work proposes SchemaRAG, a schema-guided Retrieval-Augmented Generation framework that adapts to the requirements determined from the query and the set of documents. SchemaRAG has been designed to improve knowledge-intensive reasoning tasks in Large Language Models. In contrast to previous works that used fixed templates like graphs and tables to represent queries and documents for indexing and generation purposes, the proposed SchemaRAG adapts the construction of the composite knowledge schema depending on the query and the set of documents. Formula-based indices are used as the knowledge structure. SchemaRAG consists of three main components: Schema Thinking, Schema Construction, and Schema Utilization. Experiments conducted in the Loong benchmark set, involving the legal, financial, and academia sectors, reveal the effectiveness of the proposed SchemaRAG compared to other competitive methods.

**Strengths:**

1. In contrast to previous methods that considered only a limited set of structural choices, the proposed scheme assembles the structures of knowledge and indexes based upon the requirements of each query.

2. Table 3's ablation study highlights the role of schema thinking, construction, and usage. In Figures 4-6, the visualization of schema distributions and case studies increases the clarity of the results.

3. Figures 1 and 2 are highly informative in contrasting SchemaRAG’s adaptability against traditional, rigid methods and in depicting the pipeline of the proposed system.

**Weaknesses:**

1. There is a contradiction between Definition 1 and Equation 5. Definition 1 implies U is built from a single document D. However, Equation 5 in Section 3.2 states U=K(D^1,D^2,...,D^n), which suggests the entire document collection is passed to the operator K to produce a single document's structure U. U^i appear derived from all documents, rather than per-document (U^i=K(D^i), expected).

2. The entire framework's performance hinges on a predefined candidate set of knowledge structures K and index structures I. The paper does not discuss how this set was curated or how a user might extend it for a new domain with different structural properties.

3. There is no direct, structured, or objective evaluation of schema correctness or interpretability. For example, how often are constructed schemas semantically appropriate (e.g., does a table schema capture real-world relationships), and could automated metrics for schema validity be provided?

4. The cost comparison for GraphRAG (Table 4) reports a "Construction Phase" time of 215.3 minutes. This implies a per-query index construction, which is an unconventional and highly inefficient way to use a graph index (which is typically pre-computed over the corpus). If it is pre-computation, it should not be included in the per-query "Total Phase" time. This ambiguity makes the cost-benefit analysis versus GraphRAG difficult to interpret.

5. The authors only compare SchemaRAG against original GraphRAG baselines. Several key and highly relevant baselines [1-6] are missing from the main results table (Table 1), which makes it difficult to assess the true contribution of SchemaRAG.

[1] HippoRAG2: From RAG to Memory: Non-Parametric Continual Learning for Large Language Models

[2] ArchRAG: Attributed Community-based Hierarchical Retrieval-Augmented Generation

[3] KET-RAG: A Cost-Efficient Multi-Granular Indexing Framework for Graph-RAG

[4] PIKE-RAG: sPecIalized KnowledgE and Rationale Augmented Generation

[5] KAG: Boosting LLMs in Professional Domains via Knowledge Augmented Generation

[6] RAPTOR: Recursive Abstractive Processing for Tree-Organized Retrieval

6. As the authors mention in Appendix C.2, the experiments concentrate nearly exclusively on the legal, financial, and academic domains, as given in the Loong benchmark. Experiments are not given for open-domain tasks, conversation tasks, nor multi-modal tasks. This restricts the generalizability of SchemaRAG only to highly structured domains.

7. No shared code, and this paper is not clear enough to reproduce the results.

**Questions:**

NA

---

> ### Author Response · Authors · 2025-11-25
> **Rebuttal by Authors (1/3)**
>
> Thank you for your valuable revision suggestions for our manuscript.
> We have made certain modifications and additions based on your suggestions. Moreover, to address your concerns, we provide point-to-point responses. We have summarized your review comments as [Q1] and [Q7].
>
> **[[Q1]]: The explanation of Definition 1 and Equation 5.**
> > **Comment:** There is a contradiction between Definition 1 and Equation 5. Definition 1 implies U is built from a single document D. However, Equation 5 in Section 3.2 states U=K(D^1,D^2,...,D^n), which suggests the entire document collection is passed to the operator K to produce a single document's structure U. U^i appear derived from all documents, rather than per-document (U^i=K(D^i), expected).
>
>
> Thank you for your careful observation regarding the typo. We modify formula (5) in the revised version to: $U^{(i)}=K(D^{(i)})$
>
>
> **[[Q2]]: The explanation of knowledge structures K and index structures I.**
> > **Comment:**  The entire framework's performance hinges on a predefined candidate set of knowledge structures K and index structures I. The paper does not discuss how this set was curated or how a user might extend it for a new domain with different structural properties.
>
> **Clarification of K and I:** Our proposed knowledge structure K and index structure I are not predefined elements, but rather structural forms used to integrate dense knowledge content and dependencies. Our approach does not require a pre-defined set of structures.
>
> - Knowledge Structure K: is considered a refined organization of long text content, used to remove redundant information and shorten reasoning paths.
> - Index Structure I: is used to integrate relationships between information units (e.g., chunks/documents), explicitly describing the dependencies between different knowledge units.
>
>
> **Most Existing Research:** Factual knowledge is typically stored as triplets or organized into graph structures representing relationships between information units. This can be seen as a specific form of our knowledge schema (i.e., a combination of correlation graph and graph).
>
> Flat and long contextual information, processed by two structured operators, yields a knowledge schema S for problem-solving. Benefiting from LLM as a structured operator, information can be easily extracted from text content to transform it into a suitable knowledge structure, or an index structure can be built for each unit, as illustrated in the example in Figure 4.
>
>
>
>
>
>
> **[[Q3]]: The evaluation of knowledge schema.**
> > **Comment:**  There is no direct, structured, or objective evaluation of schema correctness or interpretability. For example, how often are constructed schemas semantically appropriate (e.g., does a table schema capture real-world relationships), and could automated metrics for schema validity be provided?
>
> To our knowledge, **there are currently no direct, automated benchmarks for evaluating structural quality or schema effectiveness**. Following the majority of existing work on structure-based RAG, we adopt an overall generation metric judged by the LLM evaluation as the standard.
>
> - Schema Semantic Quality: While structured knowledge generated by LLMs may occasionally deviate from the original document, our findings show that, in most cases, the schemas align with the specified format and meet task requirements. Case studies and statistics of intermediate results are provided in Appendix C.4 (Figures 4–6), and Table 5 includes human evaluation scores of intermediate results, such as built schemas, along with their correlation to LLM-generated answers.
>
> We appreciate the reviewer’s suggestion and will address this point in greater detail in the revised paper. After open-sourcing our code, we will release the complete intermediate result data for reference.
>
> Our future work will **further explore the development of relevant structural evaluation benchmarks** to further improve the reliability of framework evaluation and contribute to research in the field of structured RAGs.

---

> > ### Author Response · Authors · 2025-11-25
> > **Rebuttal by Authors (2/3)**
> >
> > **[[Q4]]: The discussion of Resource Consumption.**
> > > **Comment:**  The cost comparison for GraphRAG (Table 4) reports a "Construction Phase" time of 215.3 minutes. This implies a per-query index construction, which is an unconventional and highly inefficient way to use a graph index (which is typically pre-computed over the corpus). If it is pre-computation, it should not be included in the per-query "Total Phase" time. This ambiguity makes the cost-benefit analysis versus GraphRAG difficult to interpret.
> >
> > - Knowledge-Intensive Task Focus: It is worth noting that our research focuses on knowledge-intensive reasoning tasks. As described in Sec2.1, we use a task query and a set of task-related documents as the raw input for each inference step, and perform structured reorganization and utilization of information at inference time. **This differs from some GraphRAG methods that first construct an offline index among all documents before performing relevant corpus retrieval and inference generation.**
> >
> > - Statistical Method: To ensure comparison fairness, we adjusted GraphRAG to construct the corpus on a **query-by-query basis**, following the statistical approach and discussion of StructRAG, which is consistent with our task setup.
> >
> > Based on your suggestion, we will supplement the above reasons and strengthen the discussion in Sec.4.5.
> >
> >
> >
> >
> > **[[Q5]]: Additional Experiments on other domain.**
> > > **Comment:**  As the authors mention in Appendix C.2, the experiments concentrate nearly exclusively on the legal, financial, and academic domains, as given in the Loong benchmark. Experiments are not given for open-domain tasks, conversation tasks, nor multi-modal tasks. This restricts the generalizability of SchemaRAG only to highly structured domains.
> >
> > We adopted the Loong benchmark, following the existing strong baseline StructRAG for knowledge-intensive reasoning tasks. As described in Sec4.1, this benchmark covers question-answering tasks in three domains, four different reasoning types, and four different context lengths, and is considered a representative domain question-answering benchmark.
> >
> > Experiments on Additional Domain: Based on your suggestion, we supplemented this with a set of win-rate experiments on SchemaRAG vs. baselines on the Podcast Transcripts dataset.
> >
> > The comparison was conducted across four dimensions, following the verification methods used in GraphRAG and StructRAG. The results are as follows:
> >
> > | Compared Method Pair                     | Comprehensiveness | Diversity | Empowerment | Directness | Average |
> > |------------------------------------------|-------------------|-----------|-------------|------------|---------|
> > | SchemaRAG vs. Long-context | 99                | 94        | 98          | 99         |  97.50  |
> > | SchemaRAG vs. RAG          | 74                | 85        | 66          | 75         |  75.00  |
> > | SchemaRAG vs. RAPTOR       | 70                | 85        | 50          | 66         |  67.75  |
> > | SchemaRAG vs. GraphRAG     | 68                | 77        | 42          | 55         |  60.50  |
> > | SchemaRAG vs. StructRAG    | 63                | 58        | 46          | 55         |  55.50  |
> >
> > The results show that SchemaRAG achieves the best average performance in Comprehensiveness, Diversity, Empowerment, and Directness in podcast domain.
> >
> > Based on your suggestion, we will consider further exploring more domains in future work.
> >
> > **[[Q6]]: Reproducibility.**
> > > **Comment:**  No shared code, and this paper is not clear enough to reproduce the results.
> >
> > We promise to release all the code and system prompts as soon as the paper is accepted.

---

> ### Author Response · Authors · 2025-11-25
> **Rebuttal by Authors (3/3)**
>
> **[[Q7]]: Additional experiments about some baselines.**
> > **Comment:**  The authors only compare SchemaRAG against original GraphRAG baselines. Several key and highly relevant baselines [1-6] are missing from the main results table (Table 1), which makes it difficult to assess the true contribution of SchemaRAG.
>
> Based on your suggestion, we have added relevant baseline experiments for comparison, and will update the results and citations in the manuscript.
>
> | **Method**                                | **Spot.**        | **Comp.**        | **Clus.**        | **Chain.**       | **Overall**      |
> |-------------------------------------------|------------------|------------------|------------------|------------------|------------------|
> | **Set 1 (10K-50K Tokens)**                |                  |                  |                  |                  |                  |
> | ArchRAG   |       44.90           |      34.88      |       40.50        |     62.21        |      46.64            |
> | HIPPORAG2        |    35.20      |      29.84       |     25.12     |    32.80    |       31.76     |
> | RAPTOR           |         **69.88**         |        58.92       |      61.36        |      45.62      |     60.69             |
> | **SchemaRAG (Ours)**                      | 66.88            | **77.88**        | **76.33**        | **70.65**            | **73.21**        |
> | **Set 2 (50K-100K Tokens)**               |                  |                  |                  |                  |                  |
> | ArchRAG   |      26.32     |   19.83  |   42.04    |  54.18  |      35.90       |
> | HIPPORAG2        |  38.30  |     43.22  |   55.17 |   24.40   |      37.19            |
> | RAPTOR           |     67.39      |   59.67     |   53.97    |    40.02     |        57.76          |
> | **SchemaRAG (Ours)**                      | **73.62**        | **71.30**            | **66.33**        | **69.38**            | **69.30**        |
> | **Set 3 (100K-200K Tokens)**              |                  |                  |                  |                  |                  |
> | ArchRAG   |      47.31       |   32.06   |   42.97     |     51.72    |     44.42      |
> | HIPPORAG2        |    58.44     |     45.04      |    39.70    |   28.12   |         43.93         |
> | RAPTOR           |    65.87       |     58.92    |     50.28      |     34.34    |         51.75         |
> | **SchemaRAG (Ours)**                      | **80.00**        | **75.29**            | **67.06**            | **65.00**            | **71.10**        |
> | **Set 4 (200K-250K Tokens)**              |                  |                  |                  |                  |                  |
> | ArchRAG   |     16.22            |     22.48     |      29.09      |   35.99      |   25.24      |
> | HIPPORAG2        |     38.15    |      29.11     |      35.77                | 19.25    |     31.87        |
> | RAPTOR           |     59.60      |    48.42       |    43.98       |    30.29     |       46.98           |
> | **SchemaRAG (Ours)**   | 64.23        | **65.56**        | **60.45**        | **63.00**      | **63.04**        |
>
>
>
>
> We have also provided a comparison with existing Structure-Based RAG methods, claiming the innovations of our SchemaRAG compared to other works and its different task applicability.
> | **Structure-based RAG**  | **Task**      | **Input Content**    | **Offline Indexing**       | **Inference-Time Structuring**  | **Composite Structures**   | **Adaptive Strategy Guidance** |
> |--------|----------------|------------------|------------------|------------------|------------------|------------------|
> | RAPTOR      | Summarization/Mutil-hop Reasoning |    Query+ Query-wise Documents        |     ✅         |    ❌       | Tree+Chunk  |  ❌
> | GraphRAG           |   QFS    |     Query+ All Documents       |      ✅       |        ❌     | Graph+Chunk  |  ❌
> | HippoRAG2           |   Mutil-hop Reasoning  |      Query+ All Documents      |      ✅        |      ❌    | Graph |  ❌
> | ArchRAG            | QFS/Mutil-hop Reasoning    |     Query+ All Documents       |       ✅       |        ❌    | Graph+Chunk |  ❌
> | KET-RAG            |   Mutil-hop Reasoning   |     Query+ All Documents       |       ✅       |    ❌    | Graph |  ❌
> | StructRAG          |   Knowledge-Intensive Reasoning     |      Query+ Query-wise Documents      |      ❌                 |  ✅   | Graph/Chunk/Algorithm ... |  ❌
> | **SchemaRAG (Ours)**         |       Knowledge-Intensive Reasoning      |   Query+ Query-wise Documents           |    ❌      |  ✅  |    Flexible Schema     |  ✅
>
>
>
>
> -----------------
>
> We sincerely thank you for your insightful suggestions, which have enhanced the quality of our work and provided valuable guidance. If you have further concerns of our work, we would be happy to discuss them.
>
> Thank you once again, and we look forward to your reply!
>
> Best regards,
>
> Authors.

---

> > ### Comment · Reviewer_Urh7 · 2025-11-27
> >
> > Thanks for your detailed response. However, the key issues about the quality of the schema remain. As such, I'd like to maintain my score.

---

> ### Author Response · Authors · 2025-12-02
> **Rebuttal to Final Concerns from Reviewer Urh7**
>
> **Firstly,** in response to your key concerns, we reviewed existing structure-based RAG methods, focusing on construction technology, structure types, and quality assessments of intermediate structured results.
> | **Structure-based RAG**  | **Structure Types**      | **Construction Technology**    | **Evaluating Structure Quality**       | **Providing Benchmark for Structure Evaluation**  |
> |--------|----------------|------------------|------------------|------------------|
> | RAPTOR      | Tree+Chunk |    Clustering Algorithm     |    ❌         |    ❌       |
> | GraphRAG      | KG+Chunk |    LLM Prompts + Community Detection      |       ❌       |    ❌       |
> | ArchRAG            | KG+Chunk   |   LLM Prompts  + Community Detection      |        ❌       |        ❌    |
> | HippoRAG2           |  KG |   LLM Prompts     |         ❌         |      ❌    |
> | LogicRAG [1] |  Logic Graph  |  LLM Prompts   |     ❌         |   ❌  |
> |  KAR^3-RAG [2] |  Chunk |   LLM Prompts     |     ❌       |        ❌    |
> |  KnowTrace [3] |   KG |   LLM Prompts  |     ❌       |        ❌
> | StructRAG          |    Graph/Chunk/Algorithm ...   |   LLM Prompts   |     ❌          |  ❌      |
> | **SchemaRAG (Ours)**         |    Schema    |   LLM Prompts + Adaptive Strategy        |    ❌ （But providing human evaluation）     |  ❌  |
>
> *[1] You Don’t Need Pre-built Graphs for RAG: Retrieval Augmented Generation with Adaptive Reasoning Structures.*
>
> *[2] From Complex to Atomic: Enhancing Augmented Generation via Knowledge-Aware Dual Rewriting and Reasoning.*
>
> *[3] KnowTrace: Bootstrapping Iterative Retrieval-Augmented Generation with Structured Knowledge Tracing.*
>
> In summary, most existing studies lack metrics for directly and objectively assessing the quality of constructed knowledge structures (e.g., knowledge graphs and tables). As noted in our previous response, the absence of structural evaluation benchmarks remains a challenge in the field, which we aim to address in future work.
>
> -------------
> **Secondly,** regarding your concern about schema quality, we reiterate the validations and analyses presented **in the original paper**:
> - Appendix C.4 and Table 5: Present human evaluations of three intermediate pipeline results, including schema quality, and the final answer quality, along with a correlation analysis between human and LLM evaluation scores.
> - Figure 4: Shows the statistical distribution of different structural types in schema representations.
> - Figures 3, 5, and 6: Provide case studies across various task scenarios.
> -------------
> **Finally,** for this rebuttal, we added **numerous cases** showcasing different schema types across various tasks in the [anonymous link](https://anonymous.4open.science/r/SchemaRAG_cases-58F4/),  and **key pipeline-related system prompts** in the Appendix D of our revision paper. Extensive analysis and observations confirm that **our pipeline reliably produces high-quality schemas**, enabling superior answers in knowledge-intensive reasoning scenarios.

---

### Official Review · Reviewer_9vpC · 2025-11-01

**Soundness:** 2
**Presentation:** 3
**Contribution:** 3
**Rating:** 4
**Confidence:** 4

**Summary:**

This paper introduces SchemaRAG, an adaptive schema-guided Retrieval-Augmented Generation (RAG) framework designed to enhance the knowledge-intensive reasoning capabilities of Large Language Models (LLMs). Traditional RAG approaches, especially those augmented with structured templates (such as graphs or tables), often suffer from rigidity and a lack of adaptability to domain-specific or query-specific needs, leading to missed critical dependencies or the inclusion of irrelevant information. SchemaRAG addresses this by adaptively constructing a “knowledge schema” tailored to the particular query and associated documents at inference time. This schema is built through three modules: (1) Schema Thinking, which decomposes queries into sub-problems and generates schema strategies; (2) Schema Construction, which adaptively assembles domain-appropriate composite structures organizing facts within and across documents; and (3) Schema Utilization, which employs a hierarchical retrieval-merge mechanism to aggregate and synthesize information for answer generation.

**Strengths:**

- The writing and figures in the paper are clear.
- The discussion of task-specific, especially query-specific, information in RAG is meaningful.
- Compared to existing works that focus on improving structure-data construction, the proposed approach of building a schema for each query to facilitate retrieval is novel.

**Weaknesses:**

- The main concern is the efficiency of the proposed method in real-world applications. Traditional structure-based RAG and GraphRAG incur a high cost during the initial graph construction, but after the knowledge is structured over the corpus, it rarely changes. During inference, they only need to perform retrieval and generation per query, which ensures efficiency. In contrast, SchemaRAG requires constructing knowledge schemas for each query and the overall corpus at inference time, meaning that construction, retrieval, and generation are all performed for every new query. This could introduce significant efficiency issues.

- Another concern is how to avoid error accumulation during the schema construction phase, since a new schema tailored for each query and corpus is built at inference time. In traditional structure-based RAG and GraphRAG, inference and construction are separated, allowing structured knowledge to be updated and improved using external tools after construction. This is not possible in SchemaRAG, as all components are performed during inference.

- Additionally, the baselines chosen from the GraphRAG domain in the experiments are not very representative. The paper does not compare against stronger and more efficient methods such as RAPTOR [1] and HIPPORAG2 [2], which are more competitive graph-based RAG baselines. Comparing SchemaRAG with these baselines, both in terms of accuracy and efficiency, would strengthen the paper.

[1] RAPTOR: Recursive Abstractive Processing for Tree-Organized Retrieval

[2] From RAG to Memory: Non-Parametric Continual Learning for Large Language Models

**Questions:**

As noted in the Weaknesses section.

---

> ### Author Response · Authors · 2025-11-25
> **Rebuttal by Authors (1/2)**
>
> We appreciate your positive evaluation of our paper as "clear," "meaningful," and "novel." In response to your concerns, we provide point-by-point responses summarized as [Q1] to [Q3].
>
>
> **[[Q1]]: Resource Consumption and Efficiency.**
> > **Comment:** The main concern is the efficiency of the proposed method in real-world applications. Traditional structure-based RAG and GraphRAG incur a high cost during the initial graph construction, but after the knowledge is structured over the corpus, it rarely changes. During inference, they only need to perform retrieval and generation per query, which ensures efficiency. In contrast, SchemaRAG requires constructing knowledge schemas for each query and the overall corpus at inference time, meaning that construction, retrieval, and generation are all performed for every new query. This could introduce significant efficiency issues.
>
> - Knowledge-Intensive Task Focus: It is worth noting that our research focuses on knowledge-intensive reasoning tasks. As described in Sec2.1, we use a task query and a set of task-related documents as the raw input for each inference step, and perform structured reorganization and utilization of information at inference time. **This differs from some GraphRAG methods that first construct an offline index among all documents before performing relevant corpus retrieval and inference generation.**
>
> - Statistical Method: To ensure comparison fairness, we adjusted GraphRAG to construct the corpus on a **query-by-query basis**, following the statistical approach and discussion of StructRAG, which is consistent with our task setup.
>
> Based on your suggestion, we will supplement the above reasons and strengthen the discussion in Sec.4.5.
>
>
>
>
> **[[Q2]]: How to avoid error accumulation during the schema construction phase.**
> > **Comment:** Another concern is how to avoid error accumulation during the schema construction phase, since a new schema tailored for each query and corpus is built at inference time. In traditional structure-based RAG and GraphRAG, inference and construction are separated, allowing structured knowledge to be updated and improved using external tools after construction. This is not possible in SchemaRAG, as all components are performed during inference.
>
> This is an insightful question. Our observations indicate that, in most cases, the LLM successfully generates the specified schema format without requiring manual correction or modification.
>
> Compared to most structure-based RAG methods, our pipeline is based on inference-time information reorganization and reasoning in knowledge-intensive reasoning scenarios. This setting **follows StructRAG**, unlike most designs that integrate initial retrieval and offline indexing among all documents. We have also provided a comparison with existing Structure-Based RAG methods, claiming the innovations of our SchemaRAG compared to other works and its different task applicability.
>
> | **Structure-based RAG**  | **Task**      | **Input Content**    | **Offline Indexing**       | **Inference-Time Structuring**  | **Composite Structures**   | **Adaptive Strategy Guidance** |
> |--------|----------------|------------------|------------------|------------------|------------------|------------------|
> | RAPTOR      | Summarization/Mutil-hop Reasoning |    Query+ Query-wise Documents        |     ✅         |    ❌       | Tree+Chunk  |  ❌
> | GraphRAG           |   QFS    |     Query+ All Documents       |      ✅       |        ❌     | Graph+Chunk  |  ❌
> | HippoRAG2           |   Mutil-hop Reasoning  |      Query+ All Documents      |      ✅        |      ❌    | Graph |  ❌
> | ArchRAG            | QFS/Mutil-hop Reasoning    |     Query+ All Documents       |       ✅       |        ❌    | Graph+Chunk |  ❌
> | KET-RAG            |   Mutil-hop Reasoning   |     Query+ All Documents       |       ✅       |    ❌    | Graph |  ❌
> | StructRAG          |   Knowledge-Intensive Reasoning     |      Query+ Query-wise Documents      |      ❌                 |  ✅   | Graph/Chunk/Algorithm ... |  ❌
> | **SchemaRAG (Ours)**         |       Knowledge-Intensive Reasoning      |   Query+ Query-wise Documents           |    ❌      |  ✅  |    Flexible Schema     |  ✅
>
>
> Finally, based on your suggestion, we will discuss this point in more detail in the revision.

---

> ### Author Response · Authors · 2025-11-25
> **Rebuttal by Authors (2/2)**
>
> **[[Q3]]: Additional experiments about competitive graph-based RAG baselines.**
> > **Comment:** Additionally, the baselines chosen from the GraphRAG domain in the experiments are not very representative. The paper does not compare against stronger and more efficient methods such as RAPTOR [1] and HIPPORAG2 [2], which are more competitive graph-based RAG baselines. Comparing SchemaRAG with these baselines, both in terms of accuracy and efficiency, would strengthen the paper.
>
> Based on your suggestion, we have added relevant baseline experiments for comparison, and will update the results and citations in the manuscript.
>
> | **Method**                                | **Spot.**        | **Comp.**        | **Clus.**        | **Chain.**       | **Overall**      |
> |-------------------------------------------|------------------|------------------|------------------|------------------|------------------|
> | **Set 1 (10K-50K Tokens)**                |                  |                  |                  |                  |                  |
> | ArchRAG   |       44.90           |      34.88      |       40.50        |     62.21        |      46.64            |
> | HIPPORAG2        |    35.20      |      29.84       |     25.12     |    32.80    |       31.76     |
> | RAPTOR           |         **69.88**         |        58.92       |      61.36        |      45.62      |     60.69             |
> | **SchemaRAG (Ours)**                      | 66.88            | **77.88**        | **76.33**        | **70.65**            | **73.21**        |
> | **Set 2 (50K-100K Tokens)**               |                  |                  |                  |                  |                  |
> | ArchRAG   |      26.32     |   19.83  |   42.04    |  54.18  |      35.90       |
> | HIPPORAG2        |  38.30  |     43.22  |   55.17 |   24.40   |      37.19            |
> | RAPTOR           |     67.39      |   59.67     |   53.97    |    40.02     |        57.76          |
> | **SchemaRAG (Ours)**                      | **73.62**        | **71.30**            | **66.33**        | **69.38**            | **69.30**        |
> | **Set 3 (100K-200K Tokens)**              |                  |                  |                  |                  |                  |
> | ArchRAG   |      47.31       |   32.06   |   42.97     |     51.72    |     44.42      |
> | HIPPORAG2        |    58.44     |     45.04      |    39.70    |   28.12   |         43.93         |
> | RAPTOR           |    65.87       |     58.92    |     50.28      |     34.34    |         51.75         |
> | **SchemaRAG (Ours)**                      | **80.00**        | **75.29**            | **67.06**            | **65.00**            | **71.10**        |
> | **Set 4 (200K-250K Tokens)**              |                  |                  |                  |                  |                  |
> | ArchRAG   |     16.22            |     22.48     |      29.09      |   35.99      |   25.24      |
> | HIPPORAG2        |     38.15    |      29.11     |      35.77                | 19.25    |     31.87        |
> | RAPTOR           |     59.60      |    48.42       |    43.98       |    30.29     |       46.98           |
> | **SchemaRAG (Ours)**   | **64.23**        | **65.56**        | **60.45**        | **63.00**      | **63.04**        |
>
>
>
>
> -----------------
>
> We sincerely thank you for your insightful suggestions, which have enhanced the quality of our work and provided valuable guidance for future research. If you have further concerns of our work, we would be happy to discuss them.
>
> Thank you once again, and we look forward to your reply!
>
> Best regards,
>
> Authors.

---

### Official Review · Reviewer_Vc2h · 2025-11-01

**Soundness:** 3
**Presentation:** 2
**Contribution:** 2
**Rating:** 6
**Confidence:** 4

**Summary:**

The paper proposes SchemaRAG, an adaptive schema-guided framework for improving knowledge-intensive reasoning in large language models. Unlike prior structure-based RAG methods that rely on fixed templates such as graphs or tables, SchemaRAG dynamically constructs query- and domain-specific schemas at inference time, enhancing flexibility and contextual relevance. The approach consists of three main components: Schema Thinking, which analyzes the query and retrieved documents to determine schema strategies; Schema Construction, which builds hierarchical composite structures integrating knowledge and index information; and Schema Utilization, which employs a hierarchical retrieval–merge mechanism for final answer generation. Experiments on the Loong benchmark across legal, financial, and academic domains demonstrate that SchemaRAG outperforms other baselines.

**Strengths:**

+ The paper presents SchemaRAG, a multi-stage, adaptive schema-guided framework that effectively enhances knowledge-intensive reasoning in large language models.
+ The modular pipeline design  from parsing, strategy formulation, construction to utilization,  provides flexibility and interpretability, addressing the rigidity of fixed-structure RAG methods.
+ Experimental results demonstrate strong reasoning performance and scalability to large contexts, validating the effectiveness of the proposed schema construction and utilization process.

**Weaknesses:**

- The framework’s multi-step pipeline raises concerns about error propagation across stages. It is unclear how the system handles upstream errors. For instance, if the schema planning or task parsing stage incorrectly decomposes a query, can later stages recover, or does this inevitably degrade the final reasoning outcome?
- The main drawback of SchemaRAG lies in its resource consumption. The multi-stage reasoning, construction, and hierarchical utilization processes introduce additional inference steps, leading to higher token usage and API call overhead, particularly during the utilization stage.
- Although the framework emphasizes adaptive schema generation, the “schema thinking” module does not appear to directly guide or optimize the retrieval process. For instance, if the module identifies a “tabular” schema, it could prompt the retriever to prioritize documents containing financial tables or structured data, thereby improving the efficiency and relevance of schema construction.

**Questions:**

How does the system behave when the schema generation step fails to capture the correct task structure? Could incorporating uncertainty estimation or schema verification improve the stability and interpretability of the reasoning process?

---

> ### Author Response · Authors · 2025-11-24
> **Rebuttal by Authors (1/2)**
>
> We appreciate your positive evaluation of our work and address your suggestions and concerns below.
>
> **[[Q1]]: Concerns about possible error in multi-step pipeline.**
> > **Comment:** The framework’s multi-step pipeline raises concerns about error propagation across stages. It is unclear how the system handles upstream errors. For instance, if the schema planning or task parsing stage incorrectly decomposes a query, can later stages recover, or does this inevitably degrade the final reasoning outcome?
>
> Our observations indicate that **the LLM reliably generates outputs in the specified format for the majority of cases**. While deviations in structured knowledge occasionally occur, we did not intervene or modify the model's outputs to maintain the integrity of the evaluation process. More cases, including the constructed schema and intermediate results, can be obtained in the anonymous link: [https://anonymous.4open.science/r/SchemaRAG_cases-58F4/](https://anonymous.4open.science/r/SchemaRAG_cases-58F4/).
>
> - Our Core Contribution: Our primary contribution lies in leveraging adaptive schemas to generate task-customized strategies and structured knowledge. This approach enhances the system's ability to integrate information and solve complex tasks involving dense text, as demonstrated in Table 1.
>
> - Future Work: We recognize the limitations of the current framework and are open to including a detailed discussion in the paper. Moving forward, we aim to explore reflective mechanisms and incorporate error correction strategies, particularly when interfacing with external tools, to further improve the reliability of the SchemaRAG framework.
>
>
>
>
> **[[Q2]]: Considering of retrieval process.**
> > **Comment:** Although the framework emphasizes adaptive schema generation, the “schema thinking” module does not appear to directly guide or optimize the retrieval process. For instance, if the module identifies a “tabular” schema, it could prompt the retriever to prioritize documents containing financial tables or structured data, thereby improving the efficiency and relevance of schema construction.
>
> - Our Task Setting: As detailed in Sec 2.1, we focuses on knowledge-intensive reasoning tasks. It processes task queries and retrieved task-related documents as raw input for each inference step, performing structured reorganization and information utilization during inference. **This aligns with the StructRAG task setting, unlike GraphRAG methods that build an offline document index before retrieval and inference.**
>
>
> - The Guidance of Thinking Process: The thinking process focuses on how to guide the model to construct structured knowledge based on retrieved query-wise documents and retrieve task-related sub-knowledge from the schema. For the example you mentioned, please refer to Figure 5 in our paper. The thinking process identifies the importance of table information and generates a guiding strategy. It also obtains the expected schema information during the construction phase.
>
>
> Based on your suggestion, we consider extending the pipeline to include offline indexing and pre-retrieval stages in the future, combining tool calls and RL-based agent technology to further explore thinking-based strategy generation for common retrievers.

---

> ### Author Response · Authors · 2025-11-24
> **Rebuttal by Authors (2/2)**
>
> **[[Q3]]: Resource Consumption**
> > **Comment:** The main drawback of SchemaRAG lies in its resource consumption. The multi-stage reasoning, construction, and hierarchical utilization processes introduce additional inference steps, leading to higher token usage and API call overhead, particularly during the utilization stage.
>
> - Knowledge-Intensive Task Focus:
> It is worth noting that our research focuses on knowledge-intensive reasoning tasks. As described in Sec2.1, we use a task query and a set of task-related documents as the raw input for each inference step, and perform structured reorganization and utilization of information at inference time. **This differs from some GraphRAG methods that first construct an offline index among all documents before performing relevant corpus retrieval and inference generation.**
>
> - Statistical Method: To ensure comparison fairness, we adjusted GraphRAG to construct the corpus on a **query-by-query basis**, following the statistical approach and discussion of StructRAG, which is consistent with our task setup. Detailed analysis is provided in Sec 4.5 and Appendix C.4, with additional discussions included in the limitations section.
>
>
> We have also provided a comparison with existing Structure-Based RAG methods, claiming the innovations of our SchemaRAG compared to other works and its different task applicability.
> | **Structure-based RAG**  | **Task**      | **Input Content**    | **Offline Indexing**       | **Inference-Time Structuring**  | **Composite Structures**   | **Adaptive Strategy Guidance** |
> |--------|----------------|------------------|------------------|------------------|------------------|------------------|
> | GraphRAG           |   QFS    |     Query+ All Documents       |      ✅       |        ❌     | Graph+Chunk  |  ❌
> | StructRAG          |   Knowledge-Intensive Reasoning     |      Query+ Query-wise Documents      |      ❌                 |  ✅   | Graph/Chunk/Algorithm ... |  ❌
> | **SchemaRAG (Ours)**         |       Knowledge-Intensive Reasoning      |   Query+ Query-wise Documents           |    ❌      |  ✅  |    Flexible Schema     |  ✅
>
>
> - Comparability with Baselines: While the pipeline involves multi-stage inference, our method demonstrates a relative advantage or comparable resource consumption compared to existing classic structured RAG methods within the same knowledge-intensive reasoning task setting, as shown in Table 4. Additionally, during the utilization phase, the construction of schemas and the design of structured task decomposition enable the model to solve problems progressively, moving from simple to complex sub-problems and achieving strong results. This phase also reduces token usage by 48.6% to 82.4%, while keeping the number of API calls within the same order of magnitude as baseline methods.
>
>
>
> **[[Q4]]: The improvement about schema verification.**
> > **Comment:** How does the system behave when the schema generation step fails to capture the correct task structure? Could incorporating uncertainty estimation or schema verification improve the stability and interpretability of the reasoning process?
>
> Although LLMs can generally generate the specified format, errors in output may still occur when dealing with extremely long texts or difficult tasks, such as the insights provided by Obs2 in the experiment.
>
> As you mentioned, introducing an additional verification mechanism is a very insightful idea and holds promise for further improvement. In the future, we will continue to explore introducing error correction mechanisms during inference, constructing relevant structural verification benchmarks, and exploring the combination of SFT or RL algorithms to train agents to guide the construction of better and more reliable knowledge schemas, thereby achieving better task solving.
>
> ----------
> **If you have any further concerns of our work, we would be happy to address them during the comment stage.**
>
> We look forward to your reply!
>
> Best regards,
>
> Authors.

---

### Author Response · Authors · 2025-12-03
**Global Response (1/2)**

We sincerely thank the reviewers for their time and thoughtful feedback. We are pleased with the positive recognition of our work:

$\color{red}{\text{Reviewers Vc2h}}$ give a positive rating for our work. $\color{blue}{\text{Reviewers 9vpC}}$ evaluate our work as "clear", "meaningful" and "novel". $\color{orange}{\text{Reviewers Urh7}}$ described our figures are "highly informative". $\color{green}{\text{Reviewers mGUh}}$ praised the paper for its "interpretability" and "empirical strength".

In response to the revision suggestions or concerns provided by the four reviewers, we summarized the following three key issues and proposed corresponding rebuttal or revision.



**[Resource Consumption]**: We clarified the task scenario differences between SchemaRAG and other structure-enhanced RAG methods, emphasizing that our statistical approach aligns with StructRAG. Additional details and analysis have been included in the revised Sec. 4.5.

| **Structure-based RAG**  | **Task**      | **Input Content**    | **Offline Indexing**       | **Inference-Time Structuring**  | **Composite Structures**   | **Adaptive Strategy Guidance** |
|--------|----------------|------------------|------------------|------------------|------------------|------------------|
| RAPTOR      | Summarization/Mutil-hop Reasoning |    Query+ Query-wise Documents        |     ✅         |    ❌       | Tree+Chunk  |  ❌
| GraphRAG           |   QFS    |     Query+ All Documents       |      ✅       |        ❌     | Graph+Chunk  |  ❌
| HippoRAG2           |   Mutil-hop Reasoning  |      Query+ All Documents      |      ✅        |      ❌    | Graph |  ❌
| ArchRAG            | QFS/Mutil-hop Reasoning    |     Query+ All Documents       |       ✅       |        ❌    | Graph+Chunk |  ❌
| KET-RAG            |   Mutil-hop Reasoning   |     Query+ All Documents       |       ✅       |    ❌    | Graph |  ❌
| StructRAG          |   Knowledge-Intensive Reasoning     |      Query+ Query-wise Documents      |      ❌                 |  ✅   | Graph/Chunk/Algorithm ... |  ❌
| **SchemaRAG (Ours)**         |       Knowledge-Intensive Reasoning      |   Query+ Query-wise Documents           |    ❌      |  ✅  |    Flexible Schema     |  ✅

**[Error Accumulation/Schema Quality]**:
We provided relevant analysis in the comments and included a comparative table. Compared to a large amount of previous work, LLM has demonstrated a strong ability to support the reliable construction of structural knowledge.

We have provided rich system prompts in Appendix D of the revised paper. Additionally, we shared cases with intermediate results at this anonymous link: [https://anonymous.4open.science/r/SchemaRAG_cases-58F4/](https://anonymous.4open.science/r/SchemaRAG_cases-58F4/),  to confirm the stability of our pipeline.
While this can inspire future improvements, it does not diminish the existing contributions of our work.


| **Structure-based RAG**  | **Structure Types**      | **Construction Technology**    | **Evaluating Structure Quality**       | **Providing Benchmark for Structure Evaluation**  |
|--------|----------------|------------------|------------------|------------------|
| RAPTOR      | Tree+Chunk |    Clustering Algorithm     |    ❌         |    ❌       |
| GraphRAG      | KG+Chunk |    LLM Prompts + Community Detection      |       ❌       |    ❌       |
| ArchRAG            | KG+Chunk   |   LLM Prompts  + Community Detection      |        ❌       |        ❌    |
| HippoRAG2           |  KG |   LLM Prompts     |         ❌         |      ❌    |
| LogicRAG  |  Logic Graph  |  LLM Prompts   |     ❌         |   ❌  |
|  KAR^3-RAG  |  Chunk |   LLM Prompts     |     ❌       |        ❌    |
|  KnowTrace |   KG |   LLM Prompts  |     ❌       |        ❌
| StructRAG          |    Graph/Chunk/Algorithm ...   |   LLM Prompts   |     ❌          |  ❌      |
| **SchemaRAG (Ours)**         |    Schema    |   LLM Prompts + Adaptive Strategy        |    ❌ （But providing human evaluation）     |  ❌  |

**[Additional Experiments]**:  Based on the suggestions from $\color{blue}{\text{Reviewer 9vpC}}$ and $\color{orange}{\text{Reviewer Urh7}}$, we have added strong baseline experimental results and comparative analyses in a new domain, which are now included in Table 2 and Table 7 of the revised paper.

---

> ### Author Response · Authors · 2025-12-03
> **Global Response (2/2)**
>
> In addition, we addressed the following minor issues:
>
> $\color{red}{\text{Reviewer Vc2h}}$: To address your suggestions for optimizing the retrieval process, we have re-emphasized our task setup and believe that our existing thought process mainly focuses on providing construction guidance, but can cover the pre-retrieval phase of the corpus in future work.
>
> $\color{blue}{\text{Reviewer 9vpC}}$: To address concerns about error accumulation during inference-time construction, we explained that our task follows StructRAG, avoiding pre-organized corpus indexing. We also point out that most cases involved in the method can be correctly constructed by the LLM.
>
> $\color{orange}{\text{Reviewer Urh7}}$: We addressed minor issues regarding definitions and formulas by correcting the details and adding explanations in the revised PDF. Point-to-point responses were provided for the questions Q1–Q7, which were well-received. To address the reviewer’s final concern about schema quality, we included a discussion on schema evaluation and added numerous prompts and cases in the revised version to demonstrate the quality and stability of schema construction.
>
>
> $\color{green}{\text{Reviewer mGUh}}$: We reclaim our technical contributions compared to StructRAG and others, review the existing structure-enhanced RAG techniques based on LLM prompts, and provide detailed system prompts in the revision version of the PDF to address the reviewers' final concern about the lack of release of LLM prompts.
>
>
>
>
> ----------------
>
> Due to special circumstances this year, the scope for expanded discussion has been limited. However, we are pleased to address and confirm the most critical concerns raised by reviewers $\color{orange}{\text{Urh7}}$ and $\color{green}{\text{mGUh}}$ based on their follow-up feedback.
>
> We remain committed to incorporating all reviewers' suggestions, clarifications, and additional experimental results into the revised paper. **Additionally, we look forward to integrating feedback from the AC into the final manuscript to further enhance its quality and contribution to the field.**
>
> Thank you sincerely for your time and consideration.

---

### Meta-Review · Area_Chair_89xN · 2026-01-07

**Summary:**

There are mainly 9 concerns by the four reviewers. All four reviewers concern about resource consumption. Reviewers Vc2h and 9vpC concern about error propagation across stages. Reviewers 9vpC and Urh7 concern that several key and highly relevant baselines are missing. Additionally, Reviewer Vc2h concerns that the schema thinking module does not appear to directly guide or optimize the retrieval process. Reviewer Urh7 concerns regarding definition contradiction, how this set was curated or how a user might extend it, and the lack of direct, structured, or objective evaluation of schema correctness or interpretability. Reviewer mGUh concerns about the unclear advantage over StructRAG and prompts are not provided.

**Reviewer Concerns:**

The rebuttal appears to have addressed several of the reviewer’s concerns, while a number of substantive issues remain unresolved. Specifically, concerns regarding resource consumption, missing key baselines, the claim that the schema thinking module does not guide retrieval, and an identified definition contradiction were largely addressed through additional comparisons with structure-based RAG methods, inclusion of new baseline experiments, clarification of methodological differences from prior GraphRAG-style indexing approaches, and correction of the relevant formula. In contrast, several concerns remain outstanding: the risk of error propagation across stages was not convincingly mitigated, as it relies mainly on assumptions about LLM output reliability; the explanation of how the schema set was curated or could be extended by users was considered insufficient; the work still lacks a direct and objective evaluation of schema correctness or interpretability; the claimed advantages over StructRAG have not been clearly substantiated to the reviewer’s satisfaction.

**Reviewer Scores:**

While some secondary concerns raised by the reviewers appear to have been addressed in the rebuttal, each reviewer’s core concern remains insufficiently resolved, including unresolved issues related to error propagation across stages, the curation and extensibility of the schema set, the lack of structured evaluation of schema correctness or interpretability, unclear advantages over StructRAG. As a result, it is unlikely that the reviewers would revise their original assessments even if they had fully participated in the discussion.

---

### Decision · Program_Chairs · 2026-01-26

Reject